# HSC70 coordinates COP9 signalosome and SCF ubiquitin ligase activity to enable a prompt stress response

Shunsuke Nishimura[1], Hidetaka Kioka [1✉], Shan Ding[2,3], Hideyuki Hakui [1], Haruki Shinomiya[1], Kazuya Tanabe[1], Tatsuro Hitsumoto[4], Ken Matsuoka[4], Hisakazu Kato[4], Osamu Tsukamoto [4,5], Yoshihiro Asano[1,6], Seiji Takashima[4,7], Radoslav I Enchev [2] & Yasushi Sakata [1]

## Abstract

The SCF (SKP1/CUL1/F-box protein) ubiquitin ligase complex plays a protective role against external stress, such as ultraviolet irradiation. The emergence of substrates activates SCF through neddylation, the covalent attachment of ubiquitin-like protein NEDD8 to CUL1. After substrate degradation, SCF is inactivated through deneddylation by COP9-signalosome (CSN), a solo enzyme that can deneddylate SCF. How the activity of CSN and SCF is coordinated within the cell is not fully understood. Here, we find that heat-shock cognate 70 (HSC70) chaperone coordinates SCF and CSN activation dependent on the neddylation status and substrate availability. Under basal conditions and low substrate availability, HCS70 directly enhances CSN deneddylation activity, thereby reducing SCF activity. Under SCF-activated conditions, HSC70 interacts with neddylated SCF and enhances its ubiquitination activity. The alternative interaction between HSC70 and CSN or neddylated SCF is regulated by the presence or absence of SCF substrates. The knockdown of HSC70 decreases SCF-mediated substrate ubiquitination, resulting in vulnerability against ultraviolet irradiation. Our work demonstrates the pivotal role of HSC70 in the alternative activation of CSN deneddylation and SCF substrate ubiquitination, which enables a prompt stress response.

**Keywords** HSC70; SCF Ubiquitin Ligase; COP9 Signalosome; Proteostasis; Stress Response
**Subject Categories** Post-translational Modifications & Proteolysis; Translation & Protein Quality

## Introduction

The SCF ubiquitin ligases are the founding members of cullin-RING E3 ligases (CRLs). They contain a cullin scaffold, a RING finger protein that recruits a ubiquitin-charged E2, and a substrate receptor module composed of SKP1 and one of F-box proteins (Baek et al, 2020; Duda et al, 2008; Reitsma et al, 2017; Bennett et al, 2010; Emanuele et al, 2011). F-box proteins target multiple substrates and dictate their substrate specificity. Over 60 human F-box proteins enable SCF to degrade hundreds of proteins (Skaar et al, 2009). SCF plays an essential role in governing various cellular processes including cell cycle and apoptosis (Bertoli et al, 2013; Cardozo and Pagano, 2004; Ang and Harper, 2005; Frescas and Pagano, 2008). Its activity is tightly regulated to maintain cellular homeostasis. The SCF ubiquitin ligase is mainly inactive under basal conditions where there are fewer SCF substrates; however, its activity is promptly upregulated in response to emerging SCF substrates induced by various stresses, such as ultraviolet (UV) irradiation (Nijhawan et al, 2003; Busino et al, 2003). As for activation of SCF, neddylation, the covalent attachment of NEDD8 to a conserved C-terminal domain lysine in CUL1, stimulates SCF ubiquitination activity (Duda et al, 2008). In contrast, SCF is inactivated by deneddylation, the deconjugation of NEDD8 from CUL1 by multiprotein complex, COP9 signalosome (CSN), which harbours a NEDD8 isopeptidase active site (Lingaraju et al, 2014; Cavadini et al, 2016; Mosadeghi et al, 2016; Enchev et al, 2012; Cope and Deshaies, 2003). After CSN-mediated deneddylation, SCF is susceptible to the CAND1-mediated substrate receptor exchange, then the new substrates can be recruited (Pierce et al, 2013; Liu et al, 2018; Reitsma et al, 2017; Bennett et al, 2010; Mosadeghi et al, 2016; Shaaban et al, 2023; Baek et al, 2023). Although deneddylation is the gateway for these exchange cycles, the presence of SCF substrates prevents CSN-mediated deneddylation (Enchev et al, 2012; Emberley et al, 2012; Mosadeghi et al, 2016). Therefore, the coordinated activation mechanism of CSN and SCF is vital but remains not fully elucidated.

[1]Department of Cardiovascular Medicine, Osaka University Graduate School of Medicine, Suita, Osaka, Japan. [2]The Visual Biochemistry Laboratory, The Francis Crick Institute, 1 Midland Road, London NW1 1AT, UK. [3]Department of Chemistry, King's College London, Britannia House, 7 Trinity Street, London SE1 1DB, UK. [4]Department of Medical Biochemistry, Osaka University Graduate School of Frontier Biosciences, Suita, Osaka, Japan. [5]Department of Biochemistry, Hyogo Medical University School of Medicine, Nishinomiya, Hyogo, Japan. [6]Department of Genomic Medicine, National Cerebral and Cardiovascular Center, Osaka, Japan. [7]The Osaka Medical Research Foundation for Intractable Diseases, Sumiyoshi-ku, Osaka, Japan. ✉E-mail: kioka@cardiology.med.osaka-u.ac.jp

Heat-shock protein 70 chaperones (HSP70s) function in a wide range of cellular housekeeping activities, including folding newly synthesized protein and promoting refolding of misfolded denatured protein to prevent its aggregation (Rosenzweig et al, 2019; Hipp et al, 2019; Clerico et al, 2019). The mammalian cytosol contains two types of HSP70 chaperones: the constitutive isoform (HSC70) and the stress-inducible isoform (HSP70) (Rosenzweig et al, 2019). Although HSC70 and HSP70 were thought to have similar cellular roles, substantial evidence supports the diverse functions of these two chaperones (Ryu et al, 2020). In contrast to HSP70, the stress-responsible role of HSC70 is not well elucidated.

This study found the intriguing role of HSC70 in the intracellular regulation of CSN and SCF, resulting in prompt protein degradation of SCF substrates. HSC70 has alternate but functionally related roles: directly enhancing the deneddylation activity of CSN under basal conditions with a small amount of SCF substrates, and interacting with neddylated SCF and enhancing its ubiquitination activity under SCF-activated conditions with a large amount of SCF substrates. This previously unrecognized interplay among HSC70, CSN, and SCF ubiquitin ligase orchestrates an efficient regulation of CSN and SCF to maintain cellular proteostasis.

## Results

### HSC70 interacts with CSN under basal conditions

HSP70s interact with denatured proteins in response to external stress (Schlecht et al, 2011; Rosenzweig et al, 2019). Therefore, we screened coimmunoprecipitates of endogenous HSC70 in the mouse heart tissue using mass spectrometry under unstressed conditions to identify the unrecognized interacting protein of HSC70 in degradation machinery. Eight subunits of CSN were identified as HSC70 interacting proteins (Fig. 1A). Of note, the peptide counts were high for each subunit among all the identified peptides. This suggested that HSC70 interacts with the CSN complex. This endogenous protein-protein interaction was confirmed by coimmunoprecipitations of endogenous HSC70 and CSN subunits (CSN3, CSN5, and CSN8) in human cell lines (HEK293T cells and HeLa cells) and mouse heart tissue (Figs. 1B and EV1A,B). The direct interaction between HSC70 and CSN complex was verified by in vitro pulldown assay using StrepII$^{2x}$ (Strep)-tagged CSN complex recombinantly expressed and purified from High Five cells and recombinant HSC70 protein purified from *Escherichia coli* (Fig. EV1C). HSC70 consists of a nucleotide-binding domain (NBD; 1–386 aa) and a substrate-binding domain (SBD; 390–604 aa) (Fig. 1C). The SBD is subdivided into SBDα and β regions. SBDβ contains a polypeptide binding domain (PBD; 405–437 aa) which enables HSC70 to bind its substrate (Qi et al, 2013). In vitro pulldown assay using PA-HA-tagged CSN complex purified from HEK293T cells and recombinant GST-tagged HSC70 (Fig. 1D) and immunoprecipitation from HEK293T cells transiently transfected with FLAG-tagged HSC70 mutants (Fig. 1E) revealed that the PBD of HSC70 was the binding site with CSN. Furthermore, NR peptide (NRLLLTGC, an HSC70 model substrate), competitively inhibited the interaction between HSC70 and CSN (Fig. 1F) (Xu et al, 2012). This provided further evidence that PBD of HSC70 is essential for this interaction. These

data demonstrated the direct connection between HSC70 and CSN under basal conditions.

### HSC70 enhances CSN deneddylation activity

To investigate the physiological implication of tight binding of HSC70 to CSN, we examined whether CSN deneddylation activity was altered by HSC70. In vitro deneddylation assay using recombinant GST-tagged HSC70 protein and recombinant Strep-tagged CSN complex showed that HSC70 enhanced CSN deneddylation activity on CUL1 (Fig. 2A). The addition of NR peptide inhibited HSC70-mediated enhancement of CSN deneddylation activity (Fig. 2B). To clarify whether HSC70-mediated enhancement of CSN deneddylation activity is regulated in an ATPase-dependent manner or not, we used the ATPase-defective mutant (HSC70 E175S) in the in vitro deneddylation assay (Jiang et al, 2007). HSC70 E175S interacted with CSN (Fig. 2C) and enhanced CSN deneddylation activity (Fig. 2D), suggesting that HSC70 enhances CSN deneddylation activity in an ATPase-independent manner. Next, to evaluate the effect of HSC70 knockdown on the neddylation status of CUL1, we established the stable HEK293T HSC70 knockdown cell line using lentiviral short hairpin RNA (shRNA) against *HSPA8* (encoding HSC70) (Fig. 2E). The HSC70 knockdown efficacy was ~80%. HSC70 knockdown cells showed an increase in the ratio of neddylated CUL1 to total CUL1 compared with HEK293T control cells. FLAG-HSC70 (full-length) expression restored the neddylation status of CUL1, whereas the FLAG-HSC70 mutant [Δ390–604 aa (ΔSBD)] failed to restore it (Fig. 2F). These findings collectively indicated that, at basal conditions, HSC70 enhances CSN deneddylation activity, tilting SCF towards a deneddylated state.

### Neddylation-dependent interaction between HSC70 and NEDD8-SCF under SCF-activated conditions

Since HSC70 enhances CSN deneddylation activity, we hypothesized that HSC70-CSN interaction should be decreased when SCF needs to be activated. To examine whether HSC70-CSN interaction is altered depending on the neddylation status of CUL1, we performed co-immunoprecipitation of endogenous HSC70 using neddylation activating enzyme inhibitor (MLN4924, also known as Pevonedistat, which decreases the observed level of neddylated CUL1) or CSN inhibitor (CSN5i-3, which increases the observed level of neddylated CUL1) (Soucy et al, 2009; Schlierf et al, 2016) (Fig. 3A). Under control and MLN4924 treatment conditions, HSC70 interacts with CSN, and no interaction was observed between non-neddylated CUL1 and HSC70. In contrast, CSN5i-3 treatment increased NEDD8-CUL1 and attenuated HSC70-CSN interaction compared to the conditions under control or MLN4924 treatment. In addition, we unexpectedly identified the interaction between HSC70 and NEDD8-CUL1. NEDD8-CUL1 is a component of SCF ubiquitin ligase complex. Pulldown assays using purified Strep-tagged NEDD8-CUL1/RBX1/SKP1/SKP2/CKS1 (NEDD8-SCF) full complex or NEDD8-CUL1/RBX1 subcomplex and recombinant GST-tagged HSC70 protein revealed that the NEDD8-SCF full complex directly interacted with HSC70 while the NEDD8-CUL1/RBX1 subcomplex did not (Fig. 3B). Furthermore, pulldown assay using GST-tagged SKP1/2/CKS1 mini-complex and HSC70 revealed that the SKP1/2/CKS1 mini-complex alone did not interact with HSC70 (Fig. EV1D), suggesting that fully

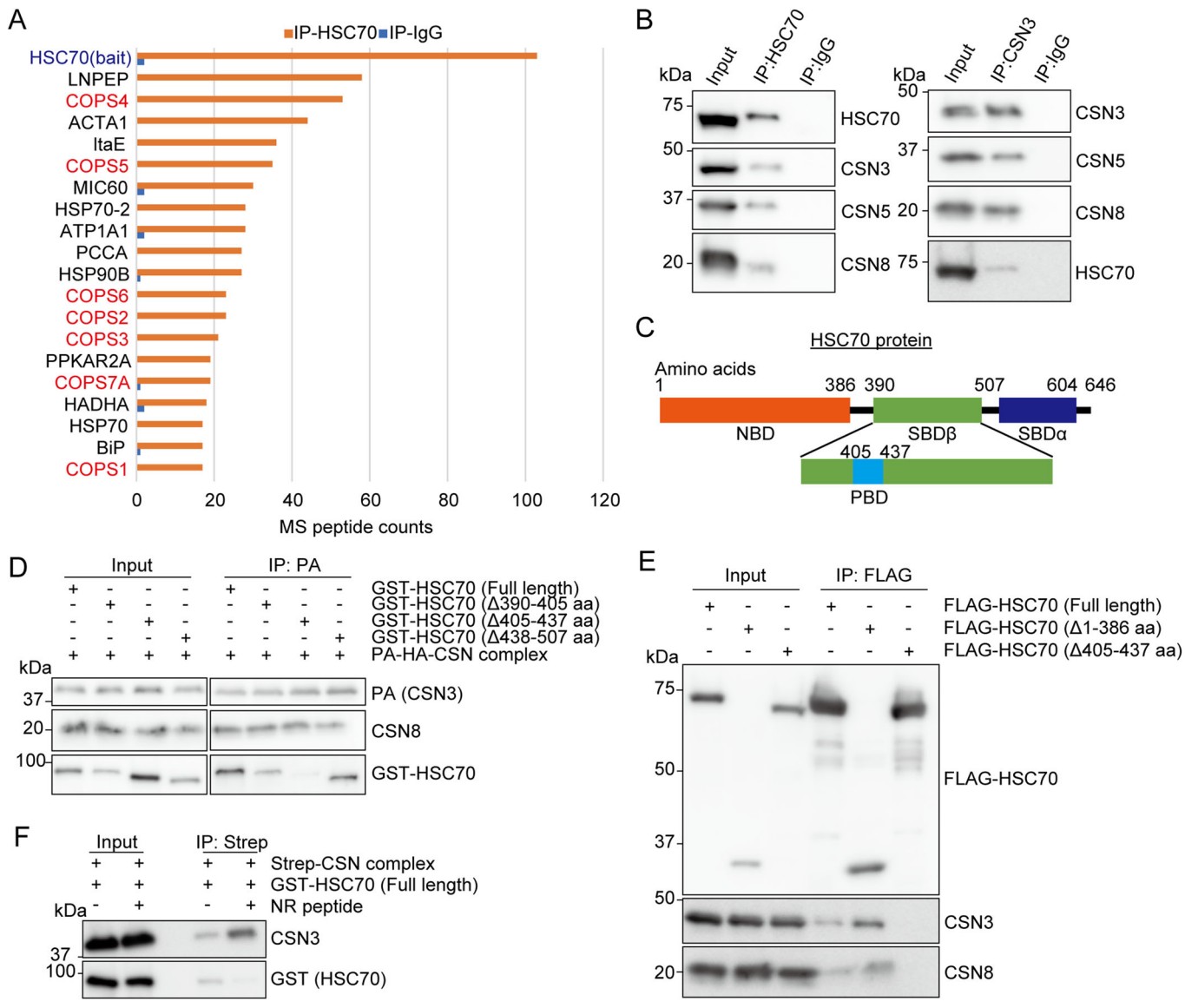

**Figure 1. HSC70 interacts with CSN complex.**

(A) Proteomic analysis of co-immunoprecipitation (co-IP) in mouse heart using HSC70 antibody (IP-HSC70), with rabbit-IgG serving as a negative control (IP-IgG). Bar graphs show mass spectrometry (MS) peptide counts of the top 20 proteins extracted from all the identified proteins (except for proteins whose peptide counts of IP-IgG was >3 to exclude non-specific binding protein). All the identified proteins are provided with the source data file. (B) Co-IP in HEK293T cells using HSC70 antibody (left) and CSN3 antibody (right), with rabbit-IgG serving as a negative control. (C) Schematic diagram of HSC70 and HSP70 protein structure. Both HSC70 and HSP70 have nucleotide binding domain (NBD) and substrate binding domain (SBD) including polypeptide binding domain (PBD). (D) In vitro pulldown assay. A mixture containing GST-tagged HSC70 mutants and purified PA-HA tagged CSN complex were subjected to immunoprecipitation with anti-PA antibody. (E) Co-IP using anti-FLAG M2 agarose in HEK293T cells transiently transfected with the indicated HSC70 mutants. (F) In vitro pulldown assay. A mixture containing Strep-tagged CSN and GST-tagged HSC70 with or without 5 μM NR peptide was subjected to immunoprecipitation with Streptactin Sepharose. Source data are available online for this figure.

assembled NEDD8-SCF could interact with HSC70, and the conformational changes of NEDD8-CUL1/RBX1 after binding to SKP1/2/CKS1 would be required for the binding to HSC70. To examine which domain of HSC70 is required for NEDD8-SCF binding, we performed in vitro GST-pulldown assay using GST-tagged HSC70 mutant proteins and Strep-tagged NEDD8-SCF full complex, which revealed NBD of HSC70 is essential for the interaction with NEDD8-SCF (Fig. 3C). These data demonstrated the neddylation-dependent direct interaction between HSC70 and neddylated SCF under SCF-activated conditions.

## HSC70 enhances the ubiquitination activity of NEDD8-SCF

To investigate the physiological implication of the binding of HSC70 to neddylated SCF, we examined the effect of HSC70 on SCF ubiquitin ligase activity with in vitro binding assays and in vitro ubiquitination assays using NEDD8-SCF and phosphor-p27 (p-p27) as a widely studied model SCF$^{SKP2}$ substrate (Frescas and Pagano, 2008; Ganoth et al, 2001). GST-pulldown assays confirmed that HSC70 formed a complex with NEDD8-SCF-p-p27,

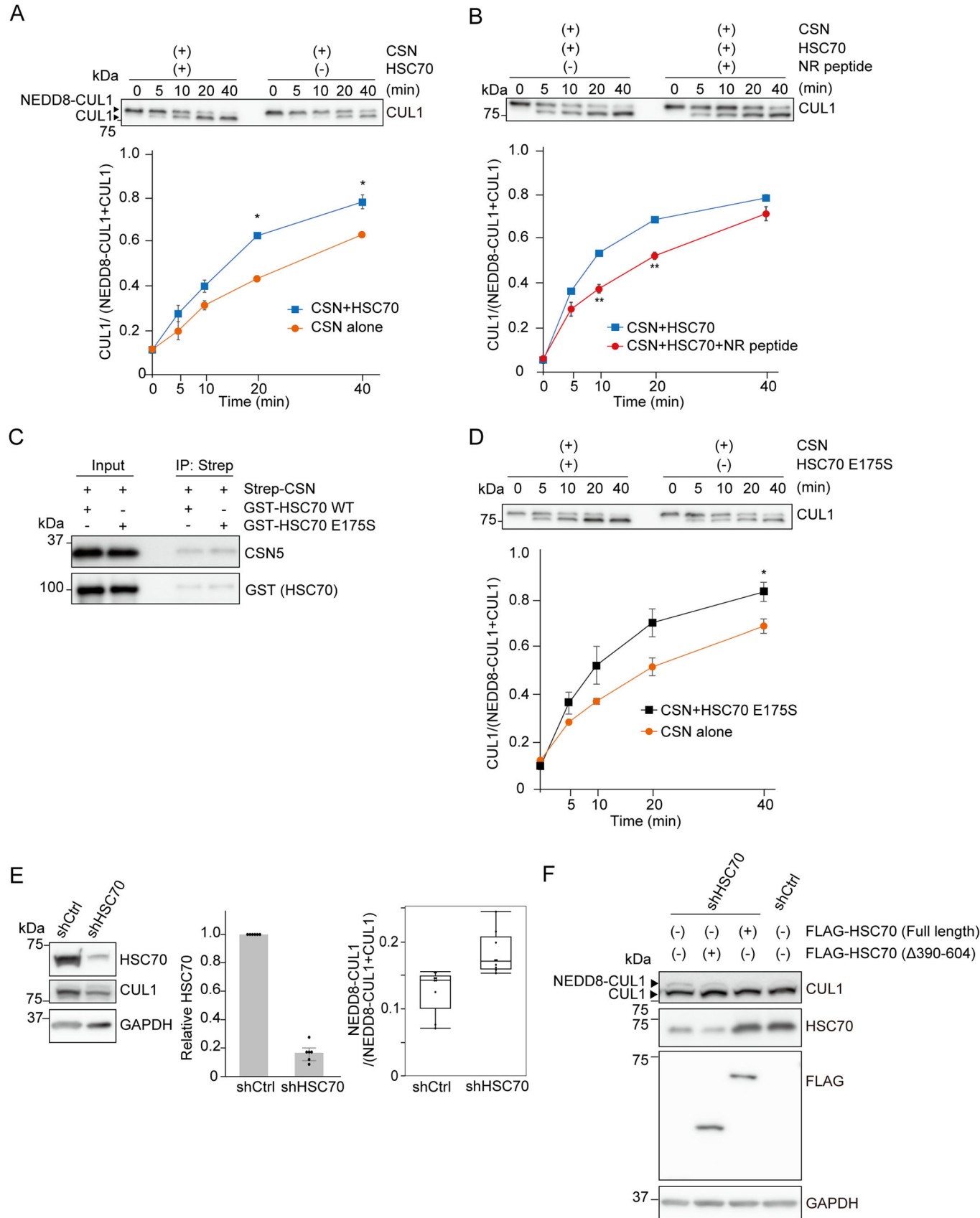

**Figure 2. HSC70 enhances the deneddylation activity of CSN.**

(A) In vitro deneddylation assay. CSN deneddylation activity in the presence or absence of 500 nM HSC70. Neddylated or non-neddylated CUL1 were detected by immunoblotting with anti-CUL1 antibody. The differences in the deneddylation activity of CSN were illustrated by estimating the percentage of non-neddylated CUL1 to total CUL1 at each time point based on protein band intensities. The bottom graphs show the mean ± s.e.m. *$P < 0.05$ ($n = 3$ independent experiments, respectively; Welch's $t$ test). A significant difference was observed at 20 min ($P = 0.014$) and 40 min ($P = 0.044$). (B) In vitro deneddylation assay. CSN deneddylation activity in the presence of 500 nM HSC70 with or without 5 μM NR peptide. The bottom graphs show the mean ± s.e.m. **$P < 0.01$ ($n = 3$ independent experiments, respectively; Welch's $t$ test). A significant difference was observed at 20 min ($P = 0.005$) and 40 min ($P = 0.0023$). (C) In vitro pulldown assay. A mixture containing Strep-tagged CSN and GST-tagged HSC70 WT or E175S mutant was subjected to immunoprecipitation with Streptactin Sepharose. (D) In vitro deneddylation assay. CSN deneddylation activity in the presence or absence of 500 nM HSC70 E175S. The bottom graphs show the mean ± s.e.m. *$P < 0.05$ ($n = 3$ independent experiments, respectively; Welch's $t$ test). A significant difference was observed at 40 min ($P = 0.036$). (E) Immunoblotting analysis of the established $HSC70$ knockdown (shHSC70) cell line. GAPDH protein was used as an internal control, and bar graphs show the relative expression level of HSC70 in shHSC70 cells to those in shCtrl cells ($n = 6$ independent expermeriments), and the ratio of neddylated CUL1 to total CUL1 in shCtrl and shHSC70 cells ($n = 9$ independent experiments). The box represents the interquartile range (IQR), with the bottom and top edges indicating the 25th and 75th percentiles, respectively. The horizontal line within the box denotes the median (50th percentile). The whiskers extend to the minimum and maximum values from the box. Data points are shown individually. (F) Immunoblotting analysis of shCtrl and shHSC70 cells with or without transient FLAG-HSC70 expression. Source data are available online for this figure.

while did not with p-p27 alone (Fig. 4A). The ubiquitination assay demonstrated that the addition of HSC70 to NEDD8-SCF (but not to non-neddylated SCF) enhanced its ubiquitination activity, while the addition of NBD deletion mutant did not (Figs. 4B and EV2A). To clarify whether HSC70-mediated enhancement of SCF ubiquitination activity is regulated in an ATPase-dependent manner, we performed a ubiquitination assay using a pharmacological inhibitor or ATPase defective mutant of HSC70. Administration of HSP70s inhibitor (YM-01) inhibited the effect of HSC70 on NEDD8-SCF mediated p-p27 ubiquitination (Fig. 4C). Furthermore, ATPase-defective mutant (HSC70 E175S) (Jiang et al, 2007) failed to enhance SCF ubiquitination activity despite interacting with NEDD8-SCF (Fig. 4D,E). It should be noted that recombinant stress-inducible HSP70 protein from *E. coli* (Fig. EV2B) did not interact with NEDD8-SCF and failed to enhance NEDD8-SCF ubiquitination activity (Fig. EV2C,D). NBD of HSC70 is essential to interact with NEDD8-SCF (Fig. 3C). We generated two chimera mutants (Fig. 4F): (i) a HSC70$_{NBD}$-HSP70$_{SBD}$ chimera mutant in which the sequence of NBD of HSP70 was changed to the corresponding sequence of HSC70 and (ii) a HSP70$_{NBD}$-HSC70$_{SBD}$ chimera mutant in which the sequence of SBD of HSP70 was changed to the corresponding sequence of HSC70. Recombinant HSC70$_{NBD}$-HSP70$_{SBD}$ chimera protein, not HSP70$_{NBD}$-HSC70$_{SBD}$ chimera protein, could restore the enhancement of NEDD8-SCF ubiquitination activity (Fig. 4G). This provided further evidence that NEDD8-SCF is specifically activated by HSC70, not HSP70. These findings demonstrated that HSC70 enhances SCF ubiquitination activity in an ATPase-dependent manner.

## SCF substrate regulates the interaction between HSC70 and CSN or NEDD8-SCF

HSC70 alternatively interacts with CSN or neddylated SCF depending on the SCF neddylation status. We investigated the underlying molecular mechanism by which HSC70 switches its interacting protein from CSN to neddylated SCF. Since CSN converges on neddylated SCF by acting as a sensor for neddylation (Enchev et al, 2012), we hypothesized that the interaction between HSC70 and CSN is required for the transition of HSC70 from CSN to neddylated SCF. To test this, we performed immunoprecipitation experiments using HEK293T cells transiently transfected with FLAG-tagged HSC70 wild type (WT) or the ΔPBD mutant (Fig. 5A). Under control conditions, FLAG-HSC70 WT interacted

with CSN, whereas the ΔPBD mutant did not. Upon SCF activation via CSN5i-3 treatment, FLAG-HSC70 WT shifted its interaction to NEDD8-SCF. In contrast, the ΔPBD mutant failed to bind NEDD8-SCF despite its ability to bind NEDD8-SCF in vitro (Fig. 3C). These findings suggest that the basal interaction between HSC70 and CSN is critical for enabling HSC70 to interact with NEDD8-SCF in vivo.

Since the presence of substrate determines whether CSN can bind to neddylated SCF (Mosadeghi et al, 2016; Enchev et al, 2012), we anticipated that whether the substrate could affect the molecular interaction among HSC70, CSN, and neddylated SCF. First, we examined whether the presence of substrate-bound neddylated SCF alters HSC70-CSN interaction. We performed GST-pulldown assay using GST-HSC70 and Strep-tagged CSN in the presence or absence of substrate-bound NEDD8-SCF-p-p27 complex. The addition of NEDD8-SCF-p-p27 decreased CSN binding to HSC70, and HSC70 interacted with NEDD8-SCF-p-p27 (Fig. 5B), suggesting that substrate-bound NEDD8-SCF competitively inhibited the HSC70-CSN interaction and, in turn, HSC70 interacts with substrate-bound NEDD8-SCF. We further examined whether substrate-free neddylated SCF influences the HSC70-CSN interaction. Since wild-type CSN deneddylate substrate-free neddylated SCF, we performed a GST-pulldown assay in the presence of CSN inhibitor (CSN5i-3). The addition of substrate-free NEDD8-SCF did not affect the binding between HSC70 and CSN, but rather HSC70 and CSN could both bind to substrate-free neddylated SCF (Fig. 5C). These findings indicated that the presence of substrate determines the interaction among HSC70, CSN, and neddylated SCF. In the presence of substrate, HSC70 switches its interacting protein from CSN to neddylated SCF, thereby enhancing ubiquitination activity. In the absence of substrate, HSC70, together with CSN, can bind to substrate-free neddylated SCF, thereby enhancing the deneddylation activity of CSN.

Subsequently, we examined that substrate-bound neddylated SCF indeed inhibits HSC70-CSN interaction in vivo. Since generating SCF substrates increases the level of substrate-bound neddylated SCF complexes (Reitsma et al, 2017; Liu et al, 2018), we performed co-immunoprecipitation of endogenous HSC70 from HEK293T cells with or without transient transfection with FLAG-tagged MCL-1 (Fig. 5D). MCL-1 is an anti-apoptotic BCL2 family member, and its degradation is regulated by SCF$^{FBW7}$ or $^{βTRCP}$ ubiquitin ligase (Inuzuka et al, 2011; Wu et al, 2020). As expected, the forced expression of MCL-1 led to the activation of SCF, and activated neddylated SCF and FLAG-MCL-1 interacted with

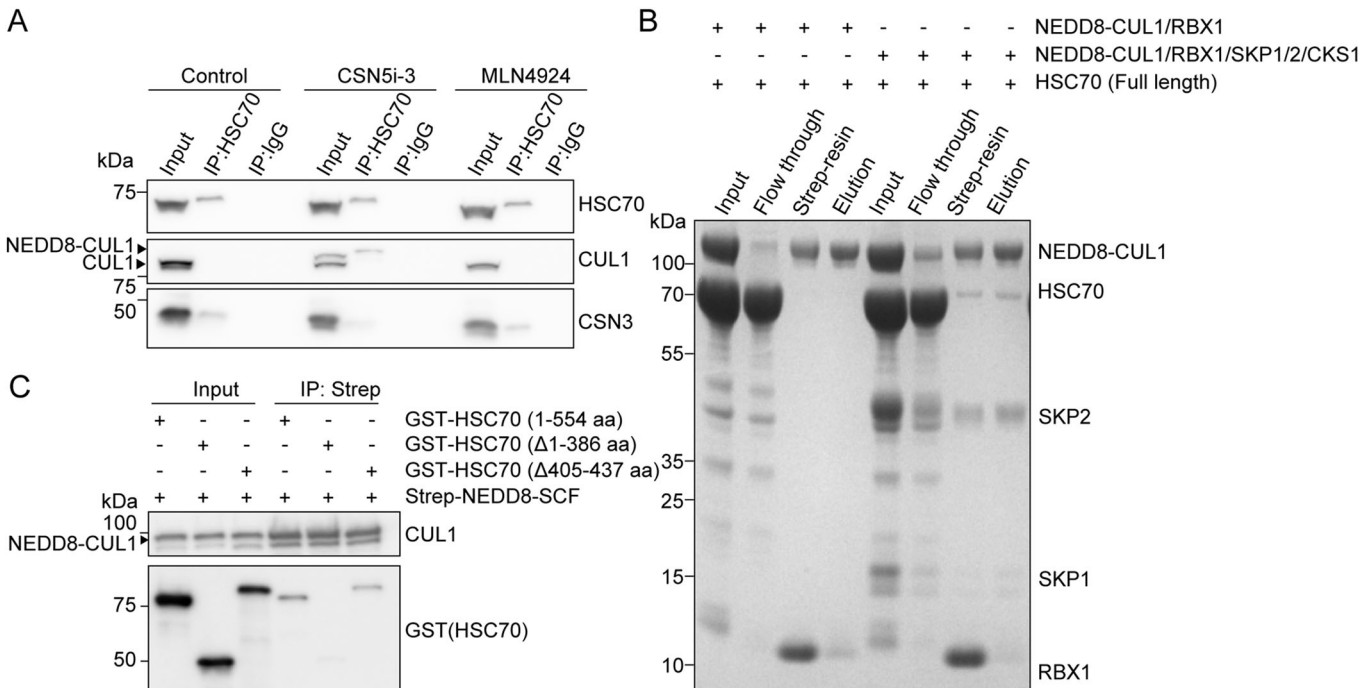

**Figure 3. HSC70 alternatively interacts with NEDD8-SCF in a neddylation-dependent manner.**

(A) Co-IP in HEK293T cells using HSC70 antibody, with IgG serving as a negative control. If indicated, pre-treatment with 1 µM MLN4924 for 10 min or 500 nM CSN5i-3 for 10 min was performed. (B) In vitro pulldown assay. HSC70 and Strep-tagged NEDD8-SCF or NEDD8-CUL1/RBX1 were subjected to immunoprecipitation with Streptactin Sepharose. The results were visualized by Coomassie brilliant blue staining. (C) In vitro pulldown assay. A mixture containing GST-tagged HSC70 mutants and Strep-tagged NEDD8-SCF was subjected to immunoprecipitation with Streptactin Sepharose. The results were visualized by immunoblotting with indicated antibodies. Source data are available online for this figure.

HSC70. On the contrary, HSC70-CSN interaction was attenuated compared with the condition without FLAG-MCL-1 expression, indicating that substrate-bound neddylated SCF competitively inhibits HSC70-CSN interaction. Furthermore, to confirm HSC70 does not bind to forced-expressed FLAG-MCL-1 non-specifically, we performed co-immunoprecipitation assay of endogenous HSC70 from HEK293T cells transiently transfected with FLAG-tagged MCL1 with or without MLN4924 treatment (Fig. 5E). Following the MLN4924 treatment, SCF was deactivated, resulting not only in the restoration of the binding between HSC70 and CSN but also in the dissipation of the binding between HSC70 and FLAG-MCL1, suggesting that neddylation as well as the presence of substrates are required for the interaction between HSC70 and SCF. In other words, only upon SCF activation, HSC70 interacts with neddylated SCF and its substrate. Without SCF activation, HSC70 cannot bind to non-neddylated SCF and FLAG-MCL1 non-specifically. These findings collectively indicate a neddylation and substrate-dependent alternative interaction between HSC70 and CSN or neddylated SCF.

## Effect of HSC70 on SCF substrate degradation in living cells

To investigate the physiological role of HSC70 in CSN-SCF regulation, we examined the protein degradation of SCF substrates. We confirmed that proteasome inhibition stabilized the amount of MCL-1 protein level (Fig. EV3A). Cycloheximide (CHX) chase

assays revealed that pharmacological inhibition of HSP70s with YM-01 or VER155008 (Ambrose and Chapman, 2021) significantly prolonged the half-life of MCL-1 (Fig. 6A), while PES-CI (a specific inhibitor of stress-inducible isoform HSP70 (Leu et al, 2009)) did not (Fig. EV3B). These results were consistent with those of in vitro ubiquitination assays performed using HSP70s inhibitor YM-01 (Fig. 4C) or recombinant HSP70 protein (Fig. EV2D). The inhibition of heat-shock protein 90 chaperones (HSP90s) with STA9090 (Ying et al, 2012) did not affect MCL-1 half-life (Fig. EV3C). These data suggested that the half-life of MCL-1 was specifically regulated by HSC70 among molecular chaperones. Furthermore, *HSC70* knockdown significantly prolonged the half-life of MCL-1 (Fig. 6B). This was validated in other *HSC70* knockdown cell lines (Fig. EV3D). Pharmacological inhibition of CSN with CSN5i-3 prolonged the half-life of MCL-1 in control cells to comparable levels in knockdown cells (Fig. EV3E), suggesting that the observed difference in MCL-1 protein half-life between the control and HSC70 knockdown cells under untreated conditions is attributable to the differential activity of SCF, not the effect of HSC70 knockdown on substrate stability. FLAG-HSC70 (full-length) expression restored the half-life of MCL-1 (Fig. 6C), whereas FLAG-HSC70 mutants [Δ1–386 aa (ΔNBD) or Δ403–437 aa (ΔPBD)] failed to restore it (Fig. EV3F). ATPase-defective mutant (FLAG-HSC70 E175S) did not influence the protein half-life of MCL-1 (Fig. EV3G) which was consistent with in vitro findings that HSC70 E175S cannot enhance the ubiquitination activity of SCF (Fig. 4E). The carboxyl-terminus of HSC70-

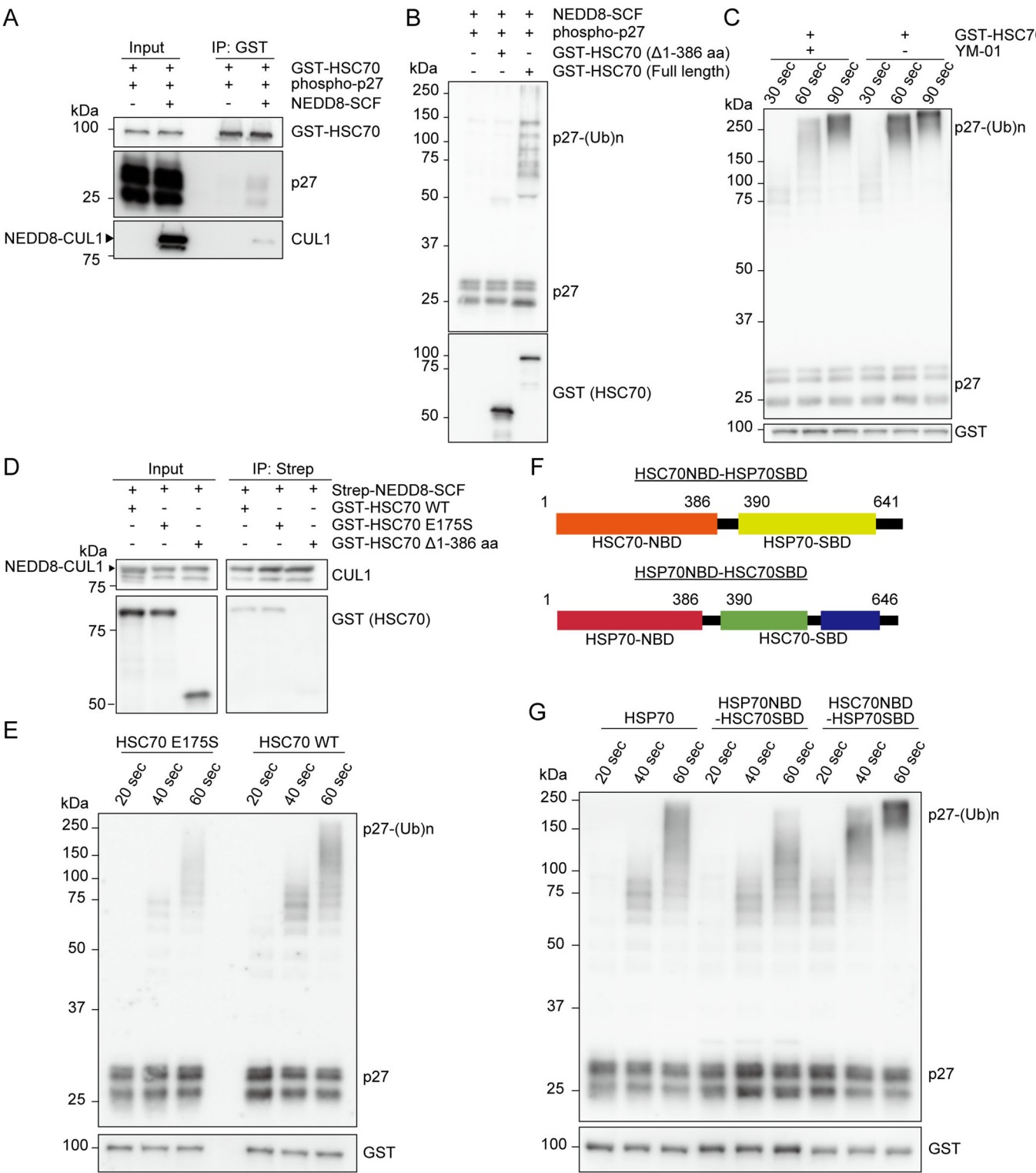

interacting protein (CHIP) is a known E3 ligase that interacts with HSC70 (Rosser et al, 2007). However, *CHIP* knockdown did not affect MCL-1 half-life (Fig. EV3H), suggesting CHIP-independent regulation. *HSC70* knockdown significantly prolonged the half-life of other known SCF substrates: cell division cycle 25A (CDC25A), p27 and c-MYC (Busino et al, 2003; Ganoth et al, 2001; Frescas and

Pagano, 2008; Welcker et al, 2004) (Fig. EV4A,B). Furthermore, tandem ubiquitin-binding entity 2 (TUBE2) pulldown assays revealed that HSP70s inhibitor (YM-01) and *HSC70* knockdown decreased the ubiquitination of endogenous MCL-1 (Fig. 6D) or overexpressed MCL-1 (Fig. 6E). Together, *HSC70* knockdown decreased ubiquitination of SCF substrates and prolonged the half-

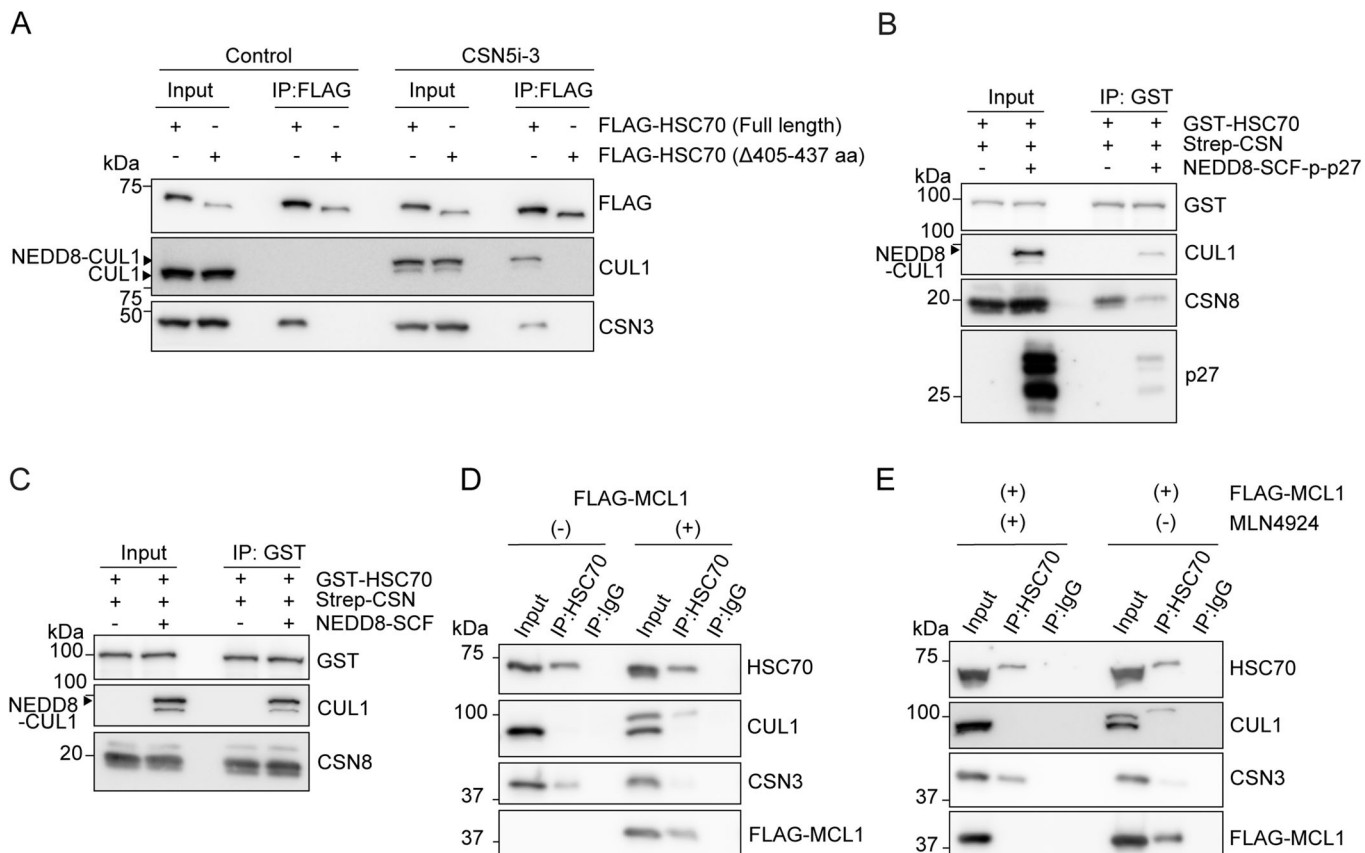

**Figure 4. HSC70 enhances the ubiquitination activity of NEDD8-SCF in an ATPase-dependent manner.**

(A) In vitro GST pulldown assay. A mixture containing GST-tagged HSC70 mutants and phospho-p27 (p-p27) with or without Strep-tagged NEDD8-SCF was subjected to immunoprecipitation. (B) NEDD8-SCF mediated in vitro ubiquitination of p-p27 assayed in the absence or presence of indicated GST-tagged HSC70 recombinant proteins. (C) NEDD8-SCF mediated in vitro ubiquitination of p-p27 assayed in the absence or presence of the HSP70s inhibitor YM-01. (D) In vitro pulldown assay. A mixture containing GST-tagged HSC70 mutants and Strep-tagged NEDD8-SCF complex was subjected to immunoprecipitation with Streptactin Sepharose. (E) In vitro ubiquitination of p-p27 assayed in the presence of indicated recombinant proteins. (F) Schematic diagram of chimera protein structure. (G) NEDD8-SCF mediated in vitro ubiquitination of p-p27 assayed in the GST-HSP70, GST-HSP70$_{NBD}$-HSC70$_{SBD}$ chimera protein or GST-HSC70$_{NBD}$-HSP70$_{SBD}$ chimera protein. (B, C, E, G) Unmodified and polyubiquitinated p-p27 were detected by immunoblotting with anti-p27 antibody. Source data are available online for this figure.

**Figure 5. Substrate-bound neddylated SCF inhibits the interaction between HSC70 and CSN, and alternatively interacts with HSC70.**

(A) Co-IP using anti-FLAG M2 agarose in HEK293T cells transiently transfected with the indicated HSC70 mutants. If indicated, pre-treatment with 500 nM CSN5i-3 for 10 min was performed. (B, C) In vitro pulldown assay. A mixture containing GST-tagged HSC70 mutants, Strep-tagged CSN and NEDD8-SCF-phospho-p27(p-p27) complex (B) or NEDD8-SCF (substrate-free) (C) was subjected to immunoprecipitation with Glutathione Sepharose 4 Fast Flow. Since wild-type CSN deneddylate substrate-free neddylated SCF, we performed GST-pulldown assay in the presence of CSN5i-3 (C). (D) Co-IP of endogenous HSC70 in HEK293T cells transiently transfected with or without FLAG-MCL1, with IgG serving as a negative control. (E) Co-IP of endogenous HSC70 in HEK293T cells transiently transfected with FLAG-MCL1 in the presence or absence of MLN4924 treatment, with IgG serving as a negative control. Source data are available online for this figure.

life of SCF substrates, which indicates that HSC70-mediated CSN-SCF regulation is required for the efficient degradation of SCF substrates in vivo.

## HSC70-mediated CSN and SCF regulation enables a prompt stress response against UV irradiation

We evaluated the effect of HSC70-mediated CSN-SCF regulation on cellular function against UV irradiation, an SCF-activating stress (Branzei and Foiani, 2008; Nijhawan et al, 2003; Karlsson-Rosenthal and Millar, 2006). Since UV irradiation generates SCF substrates and leads to increased substrate-bound neddylated SCF, we performed co-immunoprecipitation of endogenous HSC70 upon UV irradiation. Without UV irradiation, HSC70 interacts with CSN. UV irradiation increased NEDD8-CUL1 and attenuated HSC70-CSN interaction. Further, HSC70 alternatively interacted with increased NEDD8-CUL1. In contrast, MLN4924 treatment immediately after UV irradiation restored HSC70-CSN interaction (Fig. 7A), suggesting that CUL1 neddylation as well as the emergence of SCF substrates are required for the interaction

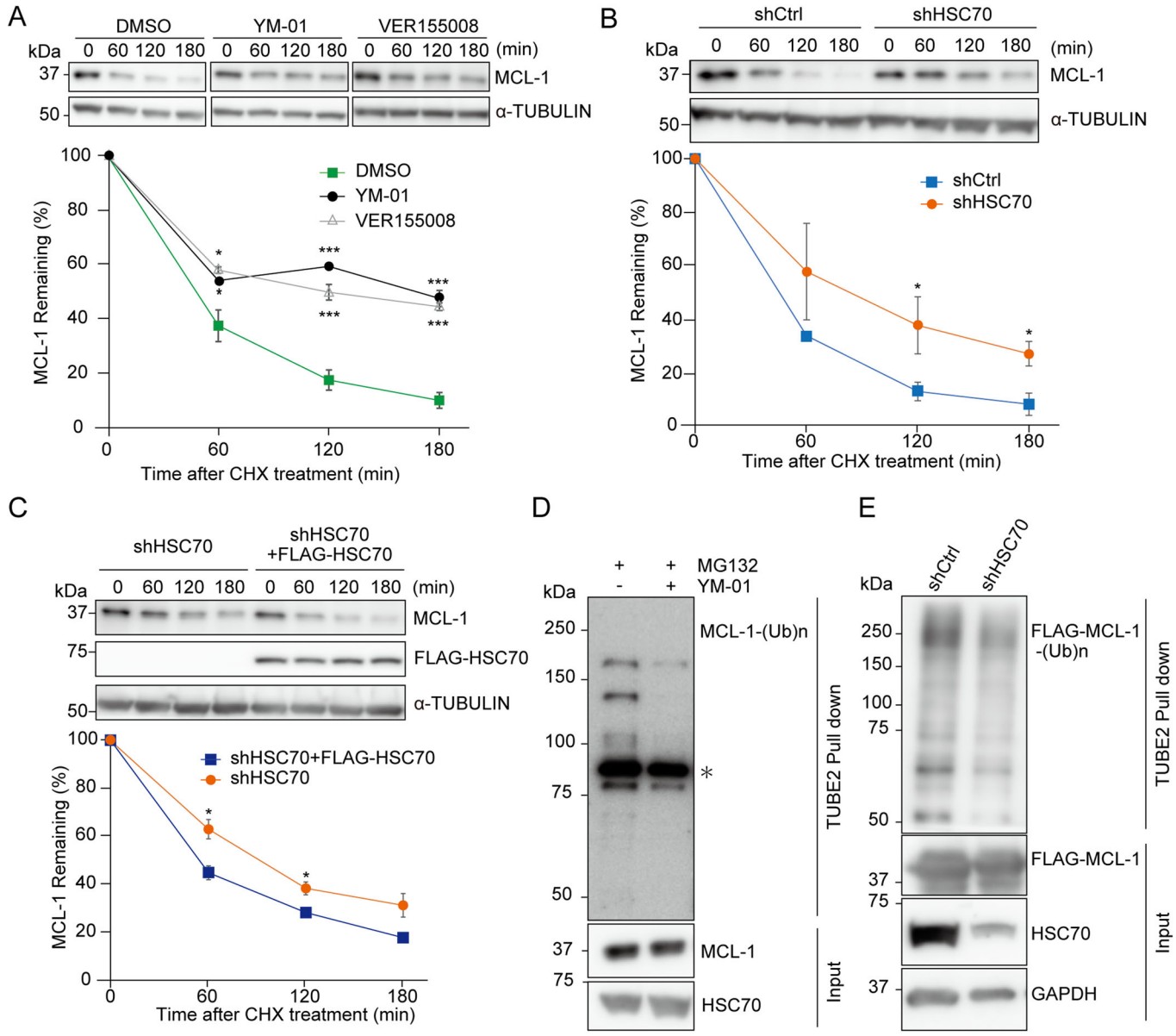

**Figure 6. Pharmacological or genetic inhibition of HSC70 decreased the ubiquitination of SCF substrates and prolonged the half-life of SCF substrates in vivo.**

(A) CHX chase assay of endogenous MCL1 with pharmacological inhibition with YM-01 or VER155008. Each protein level was densitometrically quantified (normalized to 0 min) and shown in the graph below. The bottom graph shows the mean ± s.e.m. *$P < 0.05$, ***$P < 0.001$ ($n = 3$ independent experiments, respectively; Tukey's HSD test.). Compared with DMSO treatment, a significant difference was observed at 60 min ($P = 0.014$), 120 min ($P = 0.0004$), and 180 min ($P = 0.0002$) in VER155008 treatment group, and was observed at 60 min ($P = 0.033$), 120 min ($P < 0.000087$), and 180 min ($P = 0.0001$) in YM-01 treatment group. (B) CHX chase assays of endogenous MCL-1 with or without knockdown of HSC70. The bottom graph shows the mean ± s.e.m. *$P < 0.05$ ($n = 3$ independent experiments, shCtrl; $n = 4$ independent experiments, shHSC70; Welch's $t$ test). A significant difference was observed at 120 min ($P = 0.039$) and 180 min ($P = 0.010$). (C) CHX chase assays of endogenous MCL-1 in shHSC70 cells with or without transient FLAG-HSC70 expression. The CHX chase assay was started 48 h after transfection. The bottom graph shows the mean ± s.e.m. *$P < 0.05$ ($n = 3$ independent experiments, respectively; Welch's $t$ test). A significant difference was observed at 60 min ($P = 0.026$) and 120 min ($P = 0.046$). (D) and (E) In vivo ubiquitination of endogenous MCL-1 (D) and FLAG-MCL-1 (E). Unmodified and polyubiquitinated FLAG-MCL-1 or endogenous MCL-1 were detected by immunoblotting with FLAG or MCL-1 antibody (*non-specific band). Source data are available online for this figure.

between HSC70 and SCF. Consistent with the results of immunoprecipitation under forced expression of FLAG-MCL1 (Fig. 5D,E), we successfully confirmed that alternative interaction between HSC70 and neddylated SCF under UV-induced SCF activated condition.

Next, we investigated the effect of HSC70 knockdown on UV-induced degradation of known SCF substrates. Low-dose UV irradiation induces DNA damage and stimulates SCF-mediated ubiquitination of CDC25A and subsequent rapid degradation, which is essential for prompt cell cycle arrest and restart after

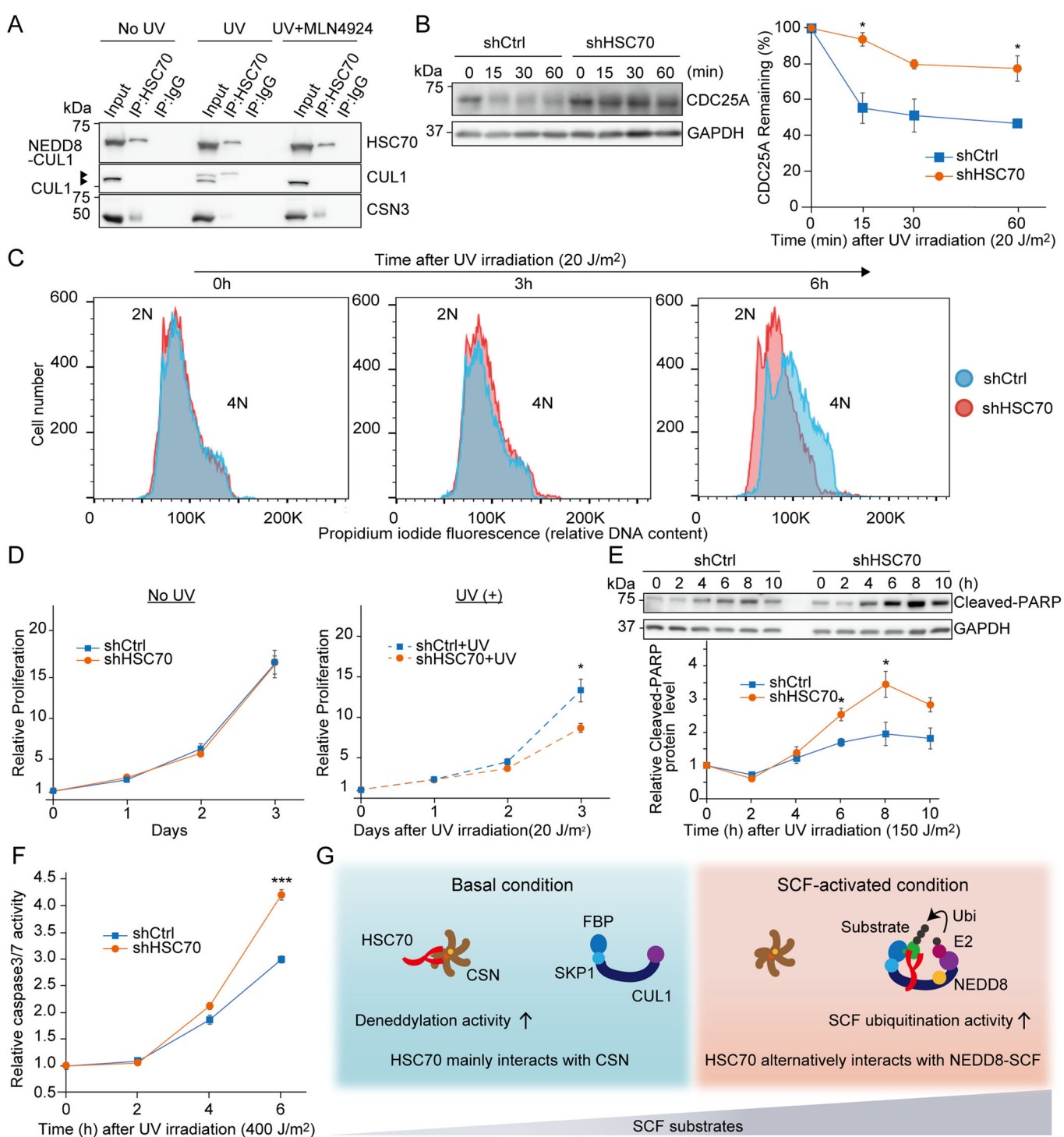

repairment of DNA damage (Karlsson-Rosenthal and Millar, 2006). We examined the effect of *HSC70* knockdown on UV-induced degradation of CDC25A and found that it impaired CDC25A degradation (Fig. 7B). We expected that delayed protein degradation of CDC25A caused by *HSC70* knockdown would affect DNA damage response to UV stress. Although immunostaining of γH2AX showed that the extent of DNA damage generated by low-dose UV irradiation (20 J/m²) was similar in control and knockdown cells (Fig. EV5A,B), cell cycle analysis using flowcytometry

revealed that *HSC70* knockdown cells showed a delayed cell cycle restart (Fig. 7C), resulting in delayed proliferation recovery (Fig. 7D). Meanwhile, *HSC70* knockdown cells showed higher levels of cleaved-poly (ADP-ribose) polymerase and caspase 3/7 than control cells under high dose UV irradiation (150 J/m² or 400 J/m²) (Fig. 7E,F). This suggested that the *HSC70* knockdown cells were vulnerable to high-dose UV irradiation. These findings collectively showed that HSC70-mediated CSN-SCF regulation is required for a prompt stress response.

**Figure 7. HSC70-mediated CSN and SCF regulation enables a prompt stress response against UV irradiation.**

(A) Co-IP in HEK293T cells with or without UV irradiation (800 J/m²) using HSC70 antibody, with IgG serving as a negative control. HEK293T cells were treated with 1 μM MLN4924 for 30 min immediately after UV irradiation where indicated. (B) Ultraviolet induced degradation of endogenous CDC25A in shCtrl and shHSC70 cells. Each protein level was densitometrically quantified (normalized to 0 min) and shown in the graph. The graphs show the mean ± s.e.m. *$P < 0.05$ ($n = 3$ independent experiments, respectively; Welch's $t$ test). A significant difference was observed at 15 min ($P = 0.031$) and 60 min ($P = 0.041$). (C) Flow cytometry assessment of the cell cycle. shCtrl or shHSC70 cells were synchronized by double thymidine block and were irradiated with low dose UV irradiation (20 J/m²) immediately after release. Cells were harvested and analyzed by fluorescence-assisted cell sorting (FACS) at the indicated time points after UV irradiation. (D) Relative proliferation rate of control or HSC70 knockdown HEK293T cells with or without UV irradiation. Relative cell counts were compared to day 0 in all graphs. The graphs show the mean ± s.e.m. *$P < 0.05$ ($n = 3$ independent experiments, respectively; Welch's $t$ test). A significant difference was observed at 3 days ($P = 0.017$) after UV irradiation. (E) Protein expression level of endogenous cleaved-PARP after UV irradiation. shCtrl or shHSC70 cells were harvested for western blotting at the indicated time point after UV irradiation. The graphs show the mean ± s.e.m. *$P < 0.05$ ($n = 3$ independent experiments, respectively; Welch's $t$ test). A significant difference was observed at 6 h ($P = 0.027$) and 8 h ($P = 0.048$) after UV irradiation. (F) Caspase3/7 assay. Relative caspase 3/7 activity to those at 0 h were shown in the graph. The graphs show the mean ± s.e.m. ***$P < 0.001$ ($n = 3$ independent experiments, respectively; Welch's $t$ test). A significant difference was observed at 6 h ($P = 0.0005$) after UV irradiation. (G) Proposed interplay among HSC70, CSN, and NEDD8-SCF. Under basal conditions, HSC70 mainly interacts with CSN and enhances its deneddylation activity. Under SCF-activated conditions, HSC70 alternatively interacts with substrate-bound NEDD8-SCF, thereby promoting SCF ubiquitination activity. Source data are available online for this figure.

## Discussion

In this study, we demonstrated that HSC70 interacts with the CSN complex and enhances its deneddylation activity under basal conditions. In contrast, HSC70 alternatively interacts with neddylated SCF and enhances its ubiquitination activity under SCF-activated conditions (Fig. 7G). We propose that this alternate but related function of HSC70 leads to an efficient SCF activation. Cells should be ready to degrade a wide range of emerging substrates. Deneddylated SCF is available for CAND1-mediated substrate receptor exchange and the recruitment of new substrates (Reitsma et al, 2017; Pierce et al, 2013; Liu et al, 2018). The promotion of SCF deneddylation by HSC70 is advantageous for cells under basal conditions. Furthermore, in the absence of deneddylation, cells are predicted to accumulate neddylated SCF for which there are no substrates. This situation leads to the autoubiquitination of substrate receptors (Petroski and Deshaies, 2005; Emberley et al, 2012; Cope and Deshaies, 2006). To avoid this situation, the deneddylation activity of CSN is needed even under basal conditions where there are few SCF substrates. On the other hand, SCF ubiquitination activity should be enhanced when SCF is activated by the emergence of SCF substrates. Our co-immunoprecipitation assay and in vitro ubiquitination assay demonstrated that HSC70 alternatively interacts with substrate-bound neddylated SCF and enhances its ubiquitination activity. Knockdown of HSC70 resulted in a reduction in CSN deneddylation and SCF ubiquitination, leading to prolonged SCF substrate degradation and vulnerability to UV stress. These findings suggest that HSC70-mediated alternative activation of CSN and SCF is required for SCF substrate degradation and prompt stress response.

Based on our findings and recent studies (Pierce et al, 2013; Reitsma et al, 2017; Enchev et al, 2012; Fischer et al, 2011; Emberley et al, 2012), we propose the following mechanism by which HSC70 switches its interacting protein from CSN to neddylated SCF. Under basal conditions with few SCF substrates available, HSC70 interacts with CSN (Fig. 1). However, when SCF substrates emerge, SCF becomes activated through neddylation and substrate binding. CSN, acting as a sensor for neddylation, converges on neddylated SCF in conjunction with HSC70. In the presence of substrates, CSN cannot bind to substrate-bound neddylated SCF. Instead, it facilitates the recruitment of HSC70 close to substrate-bound neddylated SCF. Consequently, substrate-bound neddylated SCF competitively inhibits the HSC70-CSN interaction, leading to

an alternative interaction between HSC70 and substrate-bound neddylated SCF (Fig. 5A–C). After substrate degradation and dissociation, competition arises between a new substrate and CSN for binding to substrate-free neddylated SCF. When there is a large amount of substrate, the substrate dominates this competition, promoting further substrate degradation. As the amount of substrate decreases over time, CSN begins to dominate. Substrate-free neddylated SCF can no longer inhibit the binding between HSC70 and CSN (Fig. EV5). Consequently, CSN, together with HSC70, can deneddylate substrate-free NEDD8-SCF, restoring basal conditions. In summary, HSC70 utilizes two distinct functions at the appropriate times according to the amount of SCF substrates. This could serve as a well-orchestrated mechanism to maintain cellular homeostasis through prompt protein degradation. Our findings will inspire further structural studies, opening new paths in understanding the regulation of CSN and SCF.

Regarding the difference between HSC70 and HSP70, we found that HSC70, not HSP70, enhances SCF ubiquitination activity. HSC70 and HSP70 have different expression patterns in living cells. HSC70 is the constitutively expressed form while HSP70 is highly inducible during stress (Rosenzweig et al, 2019). Since HSP70 takes some time to be expressed and cannot respond immediately to momentary stresses, a prompt stress response mechanism via HSC70 would be advantageous for cells. Furthermore, an in vitro ubiquitination assay using chimera protein revealed that the HSC70-NBD is essential for the enhancement of SCF ubiquitination activity (Fig. 3C). Via NBD, HSC70 interacts with a variety of co-chaperones, including J-domain proteins (JDPs) and nucleotide-exchange factors (NEFs) (Luengo et al, 2018; Kampinga and Craig, 2010; Rosenzweig et al, 2019; Clerico et al, 2019). Although the SCF ubiquitination activity was enhanced by HSC70 alone without the addition of other co-chaperones or JDPs in vitro, the activity may be further enhanced in cooperation with JDPs or NEFs in vivo. Although we have elucidated a previously unrecognized function of HSC70, future exploration of the factors that modify HSC70 function in vivo will also be of great importance.

Our work revealed the pivotal role of HSC70 in CSN and SCF regulation. This previously unrecognized interplay between HSC70 and CSN or SCF ubiquitin ligase provides a well-orchestrated cellular mechanism that enables prompt stress response. Failure to maintain proteostasis can cause human diseases (Hakui et al, 2022). This research extended the knowledge of molecular pathogenesis of human diseases.

# Methods

## Reagents and tools table

| Reagent/resource | Reference or source | Identifier or catalog number |
|---|---|---|
| **Experimental models** | | |
| HEK293T cells (*H. sapiens*) | ATCC | CRL-11268 |
| HeLa cells (*H. sapiens*) | ATCC | CCL2 |
| C57BL/6JJcl | CLEA Japan Inc. | N/A |
| **Recombinant DNA** | | |
| pEF-DEST51 | ThermoFisher | 12285011 |
| pLKO.1 puro | Addgene | 8453 |
| pCDH-CMV-MCS-EF1-Puro | System Biosciences | N/A |
| pRSV-Rev | Addgene | 12253 |
| pMD2.G | Addgene | 12259 |
| pMDL g/p RRE | Addgene | 12251 |
| pEF-DEST51-FLAG-HSPA8 Full length | This study | N/A |
| pEF-DEST51-FLAG-HSPA8 Δ1-386 aa | This study | N/A |
| pEF-DEST51-FLAG-HSPA8 Δ405-437 aa | This study | N/A |
| pEF-DEST51-FLAG-HSPA8 Δ390-604 aa | This study | N/A |
| pEF-DEST51-FLAG-MCL1 | This study | N/A |
| Additional information | This study (Methods) | N/A |
| **Antibodies** | | |
| Mouse anti-HSC70 | Proteintech | 66442-1(WB) |
| Rabbit anti-HSC70 | Proteintech | 10654-1(IP) |
| Rabbit anti-CSN3 | Abcam | ab79698 |
| Rabbit anti-CSN5 (JAB1) | Abcam | ab12323 |
| Rabbit anti-CSN8 | Abcam | ab124779 |
| Rabbit anti-CUL1 | Abcam | ab75817 |
| Rabbit anti-MCL-1 | Abcam | ab32087 |
| Rabbit anti-c-MYC | Abcam | ab32072 |
| Mouse anti-γH2AX | Abcam | ab26350 |
| Mouse anti-CDC25A | Santa Cruz Biotechnology | sc-7389 |
| Rabbit anti-CHIP | Cell Signaling Technology | #2080 |
| Mouse anti-p27 | Cell Signaling Technology | #3698 |
| Rabbit anti-cleaved PARP | Cell Signaling Technology | #5625 |
| Rabbit anti-α-Tubulin | MBL Life Science | PM054 |
| Mouse anti-FLAG HRP conjugated antibody | Sigma-Aldrich | A8592 |
| Mouse anti-GAPDH | Sigma-Aldrich | MAB374 |
| Rabbit anti-GST HRP conjugated antibody | Sigma-Aldrich | GERPN1236 |
| Rat anti-PA HRP conjugated antibody | Wako | 015-25951 |
| HRP-coupled sheep anti-rabbit IgG | MP Biomedicals | 12-342 |
| HRP-coupled sheep anti-mouse IgG | Rockland Immunochemicals | 610-1302 |
| Anti-PA tag antibody beads | Wako | 018-25843 |
| TUBE2 agarose | LifeSensors | UM402 |
| Anti-HA agarose (HA-7) | Sigma-Aldrich | A2095 |
| Anti-FLAG M2 agarose | Sigma-Aldrich | A2220 |
| **Oligonucleotides and other sequence-based reagents** | | |
| siCHIP sense/anti-sense | This study (Methods) | N/A |
| siControl | Sigma-Aldrich | SIC001 |
| **Chemicals, enzymes and other reagents** | | |
| MLN4924 | MedChemExpress | HY-70062 |
| CSN5i-3 | MedChemExpress | HY-112134 |
| VER155008 | MedChemExpress | HY-10941 |
| Cycloheximide | Sigma-Aldrich | C7698 |
| MG132 | Sigma-Aldrich | M7449 |
| YM-01 | Abcam | Ab146423 |
| PES-Cl | Merck | 531067 |
| STA-9090 | Selleck | S1159 |
| Recombinant human ubiquitin | R&D Systems | U-100H |
| Human p27 protein complex | Merck | 23-024 |
| Additional information | This study (Methods) | N/A |
| **Software** | | |
| FlowJo v10.6. | BD Biosciences | |
| Microsoft Excel (v1808) | Microsoft | |
| JMP Pro(v14.0.0) | JMP | |
| ImageJ v1.53t | NIH | |
| Adobe Illustrator v27.0.1 | Adobe | |
| Adobe Photoshop v24.0 | Adobe | |
| **Other** | | |
| FACSCanto™ flow cytometer | BD Biosciences | |
| Spectrolinker XL-1000 UV crosslinker | Thermo Fisher Scientific | |
| TriStar² LB 942 | Berthold Technology | |

## Reagents and antibodies

All reagents and antibodies used in this study are listed in Reagents and Tools table.

## Cell culture and siRNA transfection

HEK293T and HeLa cells were obtained from the American Type Culture Collection and cultured in Dulbecco's Modified Eagle's

medium (DMEM) with high glucose (Sigma-Aldrich), supplemented with 10% foetal bovine serum and 1% penicillin–streptomycin. Cells were confirmed negative for mycoplasma contamination using a Mycoplasma PCR Detection Kit (MycoStrip, Invitrogen). Cells were transfected with Lipofectamine 2000 (Invitrogen) and Opti-MEM (Gibco) according to the manufacturer's protocol. Knockdown of endogenous CHIP was performed by transfecting HEK293T cells with siRNA (30 nM) targeting CHIP (siCHIP, sense: 5′-CGCUGGUGGCCGUGUAUUAUU-3′; anti-sense: 5′-UAAUACACGGCCACCAGCGUU-3′) using Lipofectamine RNAiMAX (Invitrogen). siControl (SIC001, Sigma-Aldrich) was used as a negative control.

## Cloning, viral vectors, and cell lines

The coding sequences of *HSPA8* (encoding HSC70) (NM_006597.6), *CSN3* (NM_ 003653.4), and *MCL1*(NM_021960) were amplified from the human cDNA using PCR and subcloned into the Gateway pEF-DEST51 vector (Invitrogen). A FLAG tag (DYKDDDDK) was added to the N terminus of HSC70 or MCL1, and a PA tag (GVAMP-GAEDDVV) and HA tag (YPYDVPDYA) was added to the N terminus of CSN3. The HSC70 mutants were generated using PCR-based mutagenesis. A non-targeting control (SHC016, Sigma-Aldrich) (shCtrl) and shRNA against *HSPA8* (TRCN0000017279; GCAACTGTTGAAGATGAGAAA) (shHSC70) in the pLKO.1 puromycin vector (Addgene no 8453) were used, and PA-HA tagged CSN3 was cloned into the pCDH-CMV-MCS-EF1 purovector. Lentiviral particle production involved transfecting plasmids encoding shCtrl or shHSC70 in the pLKO.1 puromycin resistant vector or pCDH-CMV-PA-HA-CSN3-EF1 purovector into HEK293T cells together with pRSV-Rev (Addgene no. 12253), pMD 2.G (Addgene no. 12259), and pMDL g/p RRE plasmids (Addgene no. 12251) using Lipofectamine 2000. Cell supernatants were collected 72 h after transfection, and viral supernatants were filtered viral supernatants to remove any nonadherent packaging cells. These lentiviral particles encoding shCtrl, shHSC70, and PA-HA-CSN3 were infected into HEK293T cells and then selected with puromycin (2 µg/ml) for 2 weeks. As for HSC70 knockdown, three HSC70 knockdown cell lines were established.

## Protein purification

NEDD8-activating (APPBP1/UBA3) and conjugating enzymes (UBC12), NEDD8, SKP1/2, and CKS1 were purified as described in Mosadeghi et al (2016) (Mosadeghi et al, 2016). CUL1/RBX1, CSN complex (wild type and CSN$^{CSN5H138A}$ mutant), DEN1, and UBE1, and CDC34 were purified as described in Enchev et al (2010) and Enchev et al (2012) (Enchev et al, 2010, 2012). As for CSN$^{CSN5H138A}$ mutant purification, we modified it by skipping the HRV-3C protease cleavage to keep the His-tag on CSN5 for pulldown. CUL1/RBX1 was neddylated by incubating 8 µM CUL1/RBX1, 0.7 µM APPBP1/UBA3, 3.6 µM UBC12, and 12 µM NEDD8 in buffer containing 1.25 mM ATP, 15 mM HEPES pH 7.5, 150 mM sodium chloride, and 2.5 mM MgCl$_2$ at 37 °C for 30 min. The reaction was stopped by adding dithiothreitol (DTT) to a final concentration of 10 mM. DEN1 was used in a 1:50 ratio for 30 min at 37 °C to remove poly-neddylation. Mono-neddylated CUL1/RBX1 was purified over a Strep-tactin Superflow cartridge column (QIAGEN) and Superdex 200 column (Cytiva), which was modified

from Mosadeghi et al (2016) (Mosadeghi et al, 2016). To obtain neddylated SCF, we first prepared neddylated CUL1/RBX1, and then mixed it with SKP1/2, CKS1 with a ratio of 1:1:1. This mixture was incubated and centrifuged at 13,000 rpm for 10 min at 4 °C. The recombinant PA-HA tagged CSN complex was purified as follows: HEK293T cells stably expressing PA-HA-CSN3 protein were lysed with n-dodecyl-β-D-maltoside (DDM) buffer (20 mM NaH$_2$PO$_4$/Na$_2$HPO$_4$ [pH 7.2], 300 mM sodium chloride, 1 mM MgCl$_2$, 0.5 mM ethylenediaminetetraacetic acid (EDTA) and protease inhibitor cocktail (Nacalai-Tesque)) containing 0.25% DDM) and immunoprecipitated with anti-HA agarose for 1 h at 4 °C. The agarose was washed three times with DDM buffer, the proteins were eluted with HA peptide (Wako). Recombinant GST-HSC70 and GST-HSP70 proteins were cloned, expressed, and purified as follows: human *HSPA8* cDNA or *HSPA1* (encoding HSP70) (NM_005345) cDNA were subcloned into the pGEX-6P-1 vector. The point mutation and deletion mutant of HSC70 (E175S, Δ1–386 aa, Δ390–405 aa, Δ405–437 aa, Δ438–507 aa, and Δ555–646 aa) and chimera mutants (HSC70$_{NBD}$-HSP70$_{SBD}$ and HSP70$_{NBD}$-HSC70$_{SBD}$) were generated using PCR-based mutagenesis using wild type (WT) pGEX-6P-1/HSC70 and HSP70 as templates. pGEX-6P1/HSC70 WT, E175S, deletion mutants, HSP70 WT, HSC70$_{NBD}$-HSP70$_{SBD}$, or HSP70$_{NBD}$-HSC70$_{SBD}$ was transformed into chemically competent BL21-Rosetta *E. coli* (Invitrogen), and protein expression was induced with 0.5 mM isopropyl β-D-1-thiogalactopyranoside (Sigma-Aldrich) for 4 h at 37 °C. The cells were collected by centrifugation and lysed using B-PER buffer (Thermo Fisher Scientific). The cell lysates were agitated for 30 min at 4 °C and pulled down with Glutathione Sepharose 4 Fast Flow (Cytiva) for 1 h at 4 °C. After being washed three times, the proteins were eluted with 10 mM reduced glutathione and concentrated in DDM buffer (20 mM NaH$_2$PO$_4$/Na$_2$HPO$_4$ pH 7.2, 150 mM sodium chloride, 0.25% DDM) or HEPES buffer (15 mM HEPES pH 7.5, 150 mM sodium chloride, and 2.5 mM MgCl$_2$) using the Amicon® Ultra-15 centrifugal filter (Merck).

## Pulldown assay

Strep-tag pulldown assays were performed between Strep-CSN and full-length HSC70, Strep-NEDD8-CUL1/RBX1 and full-length HSC70, Strep-NEDD8-SCF complex and full-length HSC70 at a 1:5 ratio to Strep-Tactin beads at room temperature, and eluted with 15 mM HEPES pH 7.5, 150 mM NaCl, 2.5 mM MgCl$_2$, 0.5 mM DTT, and 2.5 mM D-desthiobiotin or sodium dodecyl sulphate-polyacrylamide gel electrophoresis (SDS-PAGE) sample buffer. For the His-tag pulldown assays, His-CSN$^{CSN5H138A}$ was pre-mixed with full-length HSC70 at a 1:5 ratio with or without NEDD8-SCF in HEPES buffer (15 mM HEPES pH 7.5, 150 mM sodium chloride, 0.5 mM TCEP). Then, the sample was loaded into Ni Sepharose High Performance resin (Cytiva). After washed with HEPES buffer twice and once with HEPES buffer plus 50 mM imidazole, the sample was eluted with HEPES buffer plus 250 mM imidazole. The results were visualized by SDS-PAGE and Coomassie brilliant blue stain. Pulldown assays between GST-HSC70 mutants and purified PA-HA tagged CSN, or between GST-HSC70 and purified His-tagged CSN with or without NEDD8-SCF-p-p27 were performed by mixing with Glutathione Sepharose 4 Fast Flow for 30 min at 4 °C. To obtain NEDD8-SCF-p-p27 complex, we first prepared phospho-p27 complex using human p27 protein complex (p27,

CyE1, and CDK2) purchased from Merck by incubating for 30 min at 30 °C in 40 mM Tris/HCl, pH 7.6, 10 mM $MgCl_2$, 1 mM ATP, and 1 mM DTT in vitro, then mixed it with NEDD8-SCF at a 3:1 ratio. Bound protein was eluted with SDS- PAGE sample buffer. The immunoprecipitants and input lysate were gel-electrophoresed and immunoblotted with the indicated antibodies.

## Deneddylation assay

In vitro deneddylation assays were performed as follows; a mixture containing 13 nM NEDD8-CUL1-RBX1 and 3 nM Strep-CSN with or without purified GST-HSC70 WT or HSC70 E175S (500 nM) was incubated in reaction buffer consisting of 20 mM Tris-HCl (pH 8.0), 200 mM sodium chloride, and 5 mM DTT. Five µM NR peptide was added if indicated. The reaction was carried out at room temperature and stopped by adding SDS-PAGE sample buffer at the indicated time points. Samples were gel electrophoresed and immunoblotted with anti-CUL1 antibody as stated above. The percentage of non-neddylated CUL1 to total CUL1 was estimated at each time point based on protein band intensities. Quantitative analyses were performed with Image J software.

## Ubiquitination assay

In vitro ubiquitination assays were performed at room temperature with 33 nM NEDD8-SCF or non-neddylated SCF, 33 nM UBE1, 166 nM CDC34, and 16.6 µM ubiquitin in the absence or presence of 60 nM GST-HSC70 recombinant protein, in the absence or presence of GST-HSP70 protein, and in the absence or presence of HSC70-HSP70 chimera mutant proteins. Ten µM HSP70s inhibitor YM-01 was added if indicated. p27 complex was phosphorylated in vitro for 30 min at 30 °C in 40 mM Tris/HCl, pH 7.6, 10 mM $MgCl_2$, 1 mM ATP, and 1 mM DTT. Twenty-six nM phospho-p27/ CyE1/CDK2 was used as substrates. The reaction was stopped by adding SDS-PAGE sample buffer. Samples were separated by SDS-PAGE and analyzed using immunoblotting. In vivo ubiquitination assays were performed as follows: HEK293T control or *HSC70* knockdown cells were transfected with a plasmid encoding FLAG-MCL-1 using Lipofectamine 2000. At 48 h after transfection, cells were treated with 10 µM proteasome inhibitor MG132 for 4 h in a $CO_2$ incubator, washed two times with PBS, and lysed with Nonidet P-40 (NP-40) buffer (10 mM Tris-HCl [pH 7.2], 150 mM sodium chloride, 1 mM EDTA, and 0.5% NP-40). Endogenous MCL-1 was detected by pre-treating cells with MG132 (10 µM) and 0.1% DMSO or 5 µM HSP70s inhibitor YM-01 for 4 h, followed by cell harvest and lysis using NP-40 buffer. These lysates were subjected to immunoprecipitation with TUBE2-agarose for 4 h at 4 °C. The agarose was washed 3 times with NP-40 buffer, then bound proteins were eluted with SDS sample buffer. The immunoprecipitants and input lysate were gel-electrophoresed and immunoblotted with the indicated antibodies.

## Ultraviolet irradiation

Cultured cells were washed two times with pre-warmed PBS and irradiated with 254 nm-ultraviolet light at the indicated dose using Spectrolinker XL-1000 UV crosslinker (Thermo Fisher Scientific). Cells were cultured in DMEM after UV irradiation.

## Caspase 3/7 assay

Caspase activity was detected using a Caspase-Glo® 3/7 assay kit (Promega). HEK293T control or *HSC70* knockdown cells were cultured at $1.5 \times 10^4$ cells per well in a 96-well plate for 24 h and irradiated with 254 nm-UV light (400 $J/m^2$). Luminescence was measured at 0, 2, and 4 h after UV irradiation using a TriStar² LB 942 (Berthold Technology) following the manufacturer's instruction. The relative caspase 3/7 activity was compared to the activity at 0 h.

## Immunoblotting

Cells were washed with PBS and lysed with NP-40 buffer (10 mM Tris-HCl pH 7.2, 150 mM sodium chloride, 1 mM EDTA, and 1.0% NP-40 substitute). The protein samples were subjected to SDS-PAGE and transferred onto polyvinylidene fluoride membranes. Membranes were incubated with the primary antibodies described above, followed by HRP-conjugated anti-rabbit or anti-mouse trueblot anti-rabbit IgG secondary antibody (MP Biomedicals and Rockland Immunochemicals). Immunoreactive signals were detected using ECL western blotting detection reagents (Cytiva).

## Animal experiments

All procedures were performed in accordance with the Guide for the Care and Use of Laboratory Animals (NIH Publication, 8th Edition, 2011), and were approved by the Osaka University Committee for Laboratory Animal Use (01-076-036). C57BL/6JJcl mice were obtained from CLEA Japan Inc. (Tokyo, Japan). All mice used in this experiment were raised under specific-pathogen-free conditions, housed under a 12-h/12-h light/dark cycle.

## Immunoprecipitation

Male mouse heart samples were immunoprecipitated with primary antibodies (anti-HSC70, anti-CSN3, or anti-CSN8), followed by Dynabeads M-280 anti-rabbit IgG (Invitrogen) for 60 min at 4 °C. Non-specific IgG (Sigma-Aldrich) and beads were used as negative controls. The beads were washed 3 times, then eluted with SDS-PAGE sample buffer. Immunoprecipitation from HEK293T cells or HeLa cells involved lysing the cells with NP-40 buffer (10 mM Tris-HCl pH 7.2, 150 mM sodium chloride, 1 mM EDTA, and 0.5% NP-40) and incubating the lysate with anti-HSC70 antibody, anti-CSN3 antibody, or anti-FLAG M2 agarose for 60 min. Beads were washed three times using NP-40 buffer before elution with SDS-PAGE sample buffer. The immunoprecipitants and input lysate were gel-electrophoresed and immunoblotted with the indicated antibodies. Cells were treated with 1 µM MLN4924 or 500 nM CSN5i-3 for 10 min before lysis with NP-40 buffer where indicated. Ultraviolet irradiation of cells was performed at 254 nm (800 $J/m^2$) at 30 min before cell lysis.

## Mass spectrometry

Immunoprecipitated beads were eluted with urea buffer (50 mM Tris-HCl pH 8.0, 150 mM sodium chloride, 10 mM EDTA, 2% sodium deoxycholate, and 6 M urea). Precipitated proteins were digested with trypsin and analyzed by LC-MS/MS using an

UltiMate 3000 Nano LC system (Thermo Fisher Scientific) coupled to a Q-Exactive mass spectrometer (Thermo Fisher Scientific). Peptides were separated using a 0.075 × 150 mm ESI-column (Thermo Fisher Scientific) with a 102-min linear gradient to 90% solvent B (acetonitrile containing 0.1% formic acid) at a flow rate of 300 nL/min. The mass spectrometer was operated in data-dependent analysis mode. Mass spectra were analyzed with Mascot Distiller v2.5 (Matrix Science), and searched against the Mus musculus UniProt protein sequence database (version February 2017). Precursor mass tolerance was set to 10 ppm. Product ions were searched with a mass tolerance of 0.01 Da. The target-decoy-based false discovery rate filter for spectra and protein identification was set to 1%. MS/MS spectral counts were used to identify proteins associated with HSC70.

## Immunocytochemistry

HEK293T control or *HSC70* knockdown cells with or without UV irradiation were fixed with 4% paraformaldehyde for 15 min at room temperature. Next, the cells were permeabilized with 0.2% Triton X-100 in PBS for 5 min and blocked with 1% bovine serum albumin for 1 h at room temperature. Then, the cells were incubated with γH2AX primary antibody overnight at 4 °C, followed by Alexa Fluor 488–labelled secondary antibodies (Invitrogen) for 1 h at room temperature. The cells were stained with 0.5 μg/ml Hoechst 33342 (Dojin Chemical, Inc.) for 30 min and the fluorescent images were recorded using a FluoView FV3000 confocal laser scanning microscope (Olympus). The number of Hoechst-positive cells and γH2AX-positive cells were counted per microscopic field. The ratios of the number of cells with >10 γH2AX foci to the Hoechst-positive cells were scored at each field.

## Cycloheximide chase assay

Cells were treated with 100 μg/ml CHX for the indicated times, lysed, and prepared for western blotting analysis. If indicated, cells were pre-treated with 0.1% DMSO, YM-01 (5 μM), VER155008 (4 μM), PES-CI (10 μM), or STA9090 (5 μM) for 3 h before CHX addition, or siRNA was transfected into cells using Lipofectamine 2000. The CHX chase assay was started 72 h after transfection. The density of the bands was quantified using Image J software and normalized using the internal control blot (GAPDH, α-TUBULIN, or HSC70). The relative protein levels of MCL-1, and CDC25A were compared to the time point at 0 min in all graphs of CHX chase experiments.

## Flow cytometry assessment of the cell cycle

After double thymidine block (two 16 h incubations in 2.5 mM thymidine (Sigma-Aldrich) with an 8 h release), HEK293T control or *HSC70* knockdown cells were exposed to 254 nm-ultraviolet light (20 J/m²). The cells were trypsinized and collected at various time points after UV irradiation. A total of $10^6$ cells were fixed in 70% ethanol at −20 °C for at least 4 h. The fixed cells were washed twice with PBS, followed by 10 mg/ml RNase A (Sigma-Aldrich) for 45 min at room temperature, then 500 μg/ml propidium iodide (Sigma-Aldrich) for 10 min at room temperature. The stained cells were analyzed with a FACSCanto™ flow cytometer (BD Biosciences) and a total of 30,000 events were collected for each sample.

## Cell proliferation assay

HEK293T control or *HSC70* knockdown cells were seeded in 96-well plates at 2500 cells per well and incubated for 24 h, 48 h, or 72 h. Cells were then stained with 0.5 μg/ml Hoechst 33342 (Dojin Chemical, Inc.) for 30 min at 37 °C. The stained nuclei were imaged using a GE Healthcare IN Cell Analyzer 6000. The number of positive nuclei was analyzed as cell counts. The relative cell counts were compared to day 0 in all graphs of the proliferation assay. Cells were exposed to 254 nm-ultraviolet light (20 J/m²) after adhesion wherever indicated.

## Data availability

For MS data, raw data, peak lists and result files have been deposited as accession No. PXD 042742 at the ProteomeXchange Consortium (Vizcaíno et al, 2014) via the jPOST partner repository (Okuda et al, 2017) as JPST002173.

The source data of this paper are collected in the following database record: biostudies:S-SCDT-10_1038-S44319-025-00376-x.

## Peer review information

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

## Acknowledgements

We thank Y Shintani and Y Nishida for helpful scientific discussions; M Kishimoto, K Shingu, and E Takata for technical assistance; Y Kurokawa, H Kawasaki, and Y Okada for secretarial support. We are grateful to the Center of Medical Innovation and Translational Research (CoMIT), Osaka University, for technical support. We would like to thank Editage (www.editage.com) for English language editing. This work was supported by grants from the Japan Agency for Medical Research and Development (AMED; grant nos. JP19ek0109412 and JP23ek0210184), the Japan Society for the Promotion of Science (JSPS KAKENHI; grant no. JP22K08155), JST FOREST Program (JPMJFR2353), the Takeda Science Foundation, the SENSHIN Medical Research Foundation, the Osaka Medical Research Foundation for Intractable Diseases, and the Japan Heart Foundation Research Grant for Dilated Cardiomyopathy. Work in the Enchev lab was supported by the Francis Crick Institute which receives its core funding from Cancer Research UK (CC2059), the UK Medical Research Council (CC2059), and the Wellcome Trust (CC2059).

## Author contributions

Shunsuke Nishimura: Conceptualization; Data curation; Investigation; Visualization; Methodology; Writing—original draft; Writing—review and editing. Hidetaka Kioka: Conceptualization; Supervision; Funding acquisition; Writing—original draft; Writing—review and editing. Shan Ding: Investigation; Visualization; Methodology; Writing—review and editing. Hideyuki Hakui: Investigation; Writing—review and editing. Haruki Shinomiya: Investigation; Writing—review and editing. Kazuya Tanabe: Investigation; Writing—review and editing. Tatsuro Hitsumoto: Investigation; Writing—review and editing. Ken Matsuoka: Supervision; Investigation; Writing—review and editing. Hisakazu Kato: Supervision; Investigation; Writing—review and editing. Osamu Tsukamoto: Supervision; Investigation; Writing—review and editing. Yoshihiro Asano: Supervision; Writing—review and editing. Seiji Takashima: Supervision; Writing—review and editing. Radoslav I Enchev: Supervision; Funding acquisition; Visualization; Methodology; Writing—review and editing. Yasushi Sakata: Supervision; Writing—review and editing.

Source data underlying figure panels in this paper may have individual authorship assigned. Where available, figure panel/source data authorship is listed in the following database record: biostudies:S-SCDT-10_1038-S44319-025-00376-x.

## Disclosure and competing interests statement

The authors declare no competing interests.

# Expanded View Figures

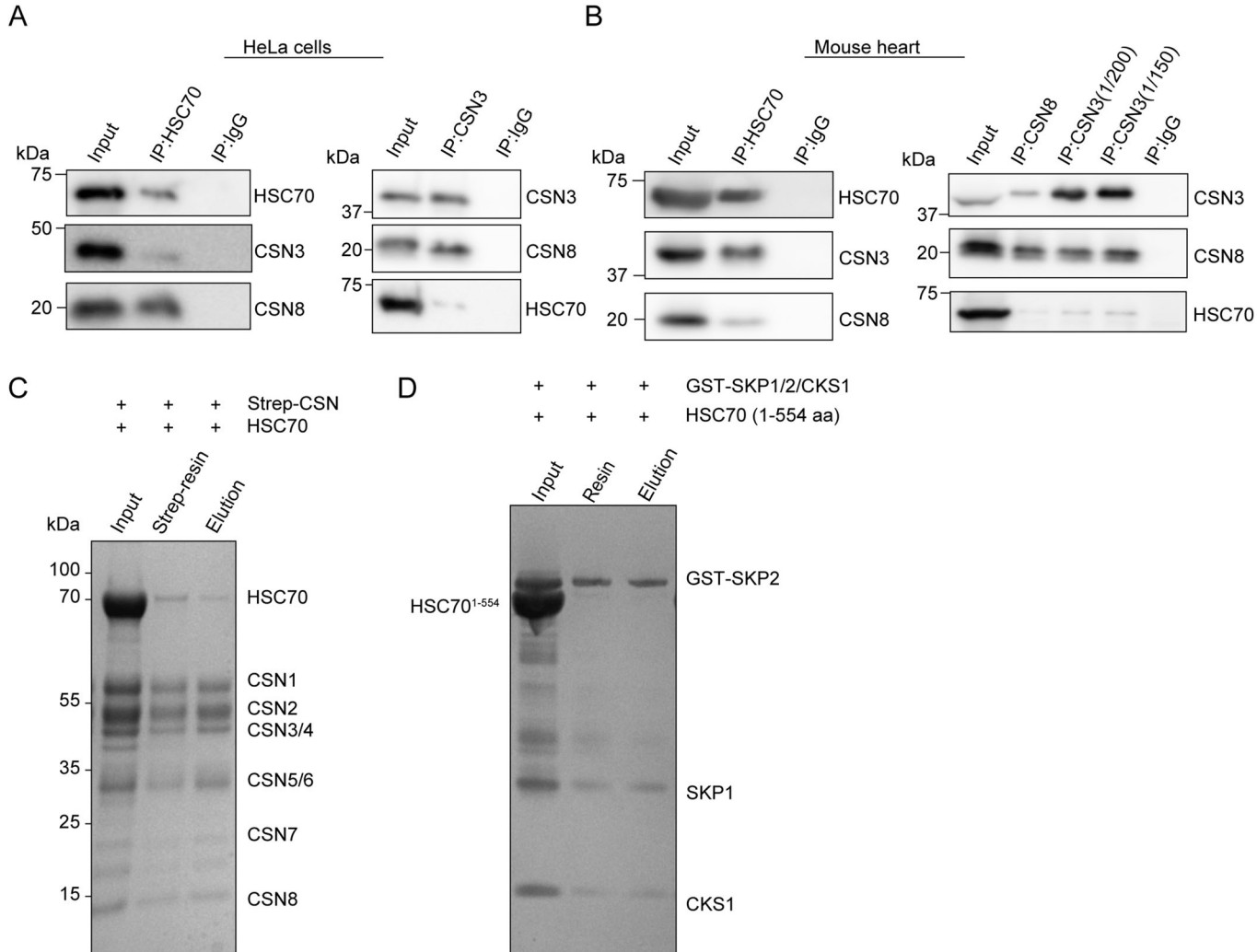

**Figure EV1. HSC70 interacts with CSN in vivo and in vitro, but does not interacts with SKP1/2/CKS mini complex.**

(A) Co-immunoprecipitation in HeLa cells using HSC70 and CSN3 antibody, with rabbit-IgG serving as a negative control. (B) Co-immunoprecipitation in mouse heart tissue using HSC70, CSN8, and CSN3 antibody (1/150 or 1/200 dilution), with rabbit-IgG serving as a negative control. (C) In vitro pulldown assay. A mixture containing strep-tagged CSN and recombinant HSC70 was subjected to immunoprecipitation with Streptactin Sepharose. The result was visualized by Coomassie brilliant blue staining. (D) In vitro pulldown assay. A mixture containing GST-tagged SKP1/2/CKS1 and HSC70[1-554aa] was subjected to immunoprecipitation with Glutathione Sepharose 4 Fast Flow. The results were visualized by Coomassie brilliant blue staining.

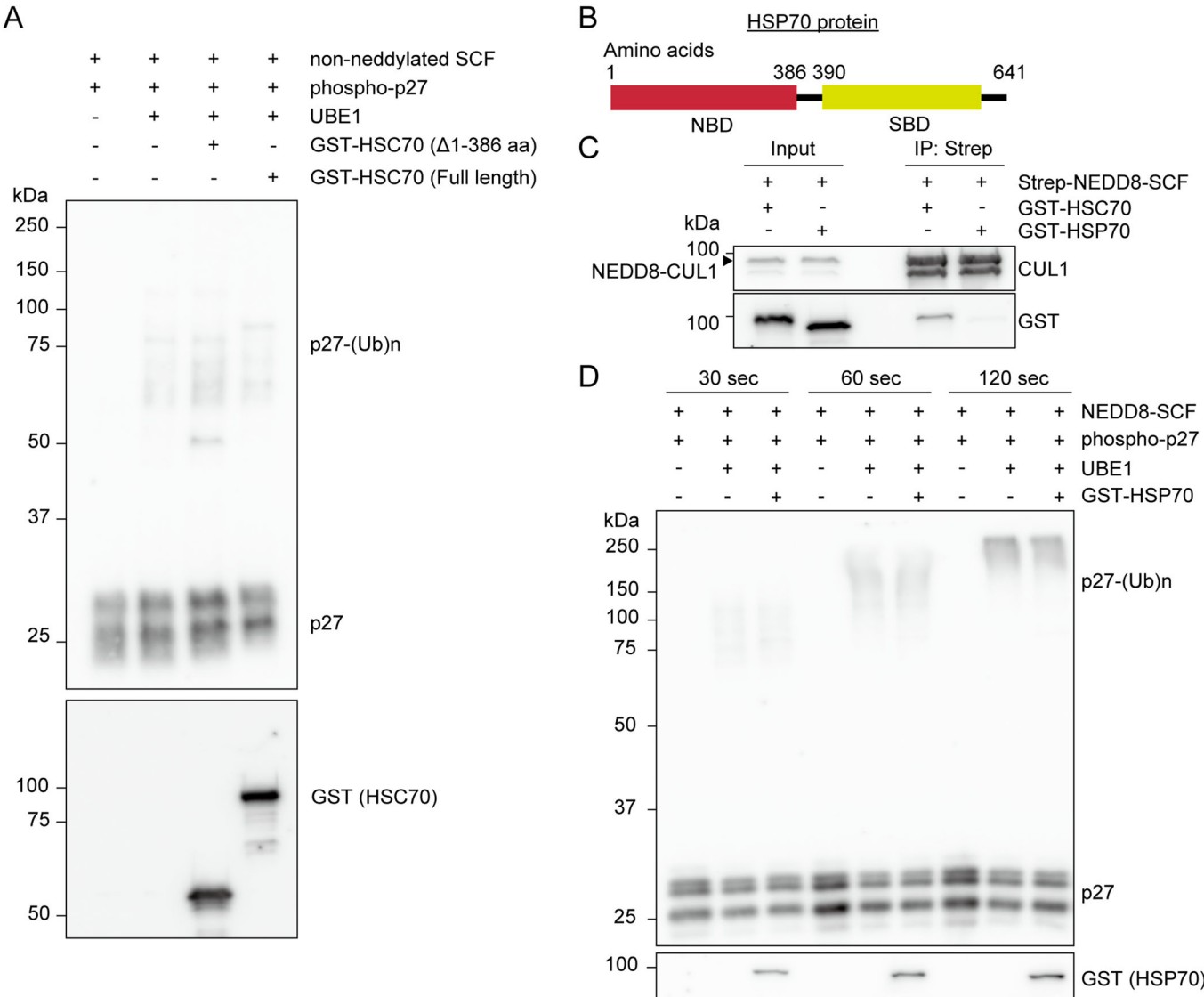

**Figure EV2. HSC70 does not enhance the ubiquitination activity of non-neddylated SCF, and HSP70 does not enhance NEDD8-SCF ubiquitination activity.**

(A) Non-neddylated SCF mediated in vitro ubiquitination of p-p27 assayed in the absence or presence of indicated recombinant proteins. Unmodified and polyubiquitinated p-p27 were detected by immunoblotting with anti-p27 antibody. (B) Schematic diagram of HSP70 protein structure. (C) In vitro pulldown assay. A mixture containing GST-tagged HSC70 or HSP70, and Strep-tagged NEDD8-SCF were subjected to immunoprecipitation with Streptactin Sepharose. (D) NEDD8-SCF mediated in vitro ubiquitination of p-p27 assayed in the absence or presence of GST-HSP70.

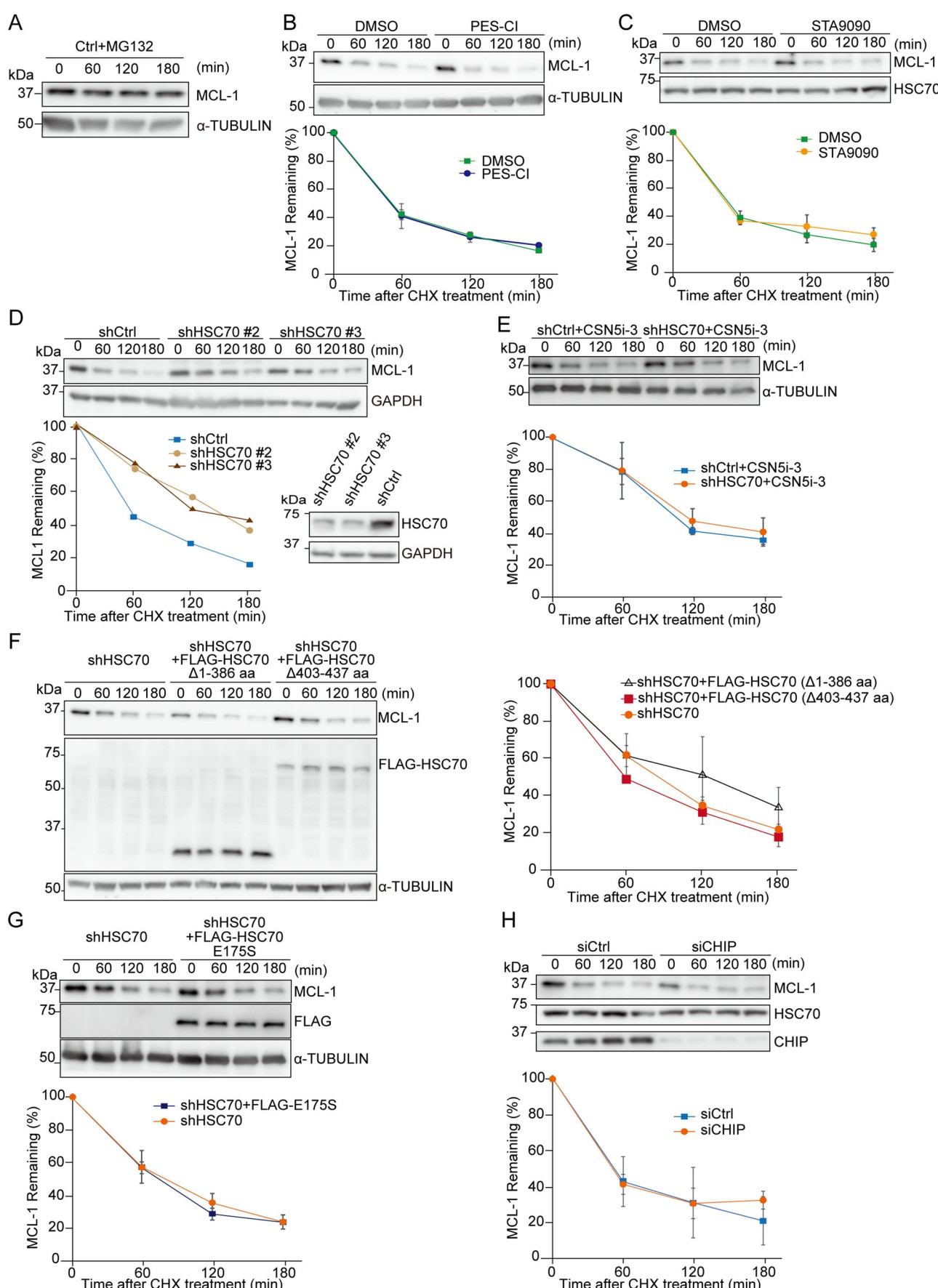

◀ **Figure EV3. Cycloheximide (CHX) chase assay in various conditions.**

(A) CHX chase assay of endogenous MCL-1 in the presence of proteasome inhibitor MG132. (B) CHX chase assay of endogenous MCL1 with pharmacological inhibition with PES-CI. The graphs show the mean ± s.e.m. ($n = 3$, respectively; Welch's $t$ test). No significant difference was observed at each time point. (C) CHX chase assay of endogenous MCL1 with pharmacological inhibition with STA9090. The graphs show the mean ± s.e.m. ($n = 3$, respectively; Welch's $t$ test). No significant difference was observed at each time point. (D) CHX chase assay of endogenous MCL-1 in other *HSC70* knockdown cell lines (shHSC70 clones #2 and 3). The lower right panel shows the extent of *HSC70* knockdown in these cell lines by immunoblotting. (E) CHX chase assay of endogenous MCL1 in *HSC70* knockdown cells with pharmacological inhibition with CSN5i-3. The graphs show the mean ± s.e.m. ($n = 3$, respectively; Welch's $t$ test). No significant difference was observed at each time point. (F) CHX chase assay of endogenous MCL-1 with or without transient FLAG-HSC70 mutant expression. The graphs show the mean ± s.e.m. (Δ1–386 aa or Δ403–437 aa; $n = 3$, respectively; Welch's $t$ test). No significant difference was observed at each time point. (G) CHX chase assay of endogenous MCL-1 with or without transient FLAG-HSC70 E175S mutant expression. The graphs show the mean ± s.e.m. ($n = 3$, respectively; Welch's $t$-test). No significant difference was observed at each time point. (H) CHX chase assay of endogenous MCL1 with or without *CHIP* knockdown. The graphs show the mean ± s.e.m. ($n = 3$, respectively; Welch's $t$ test). No significant difference was observed at each time point. (B–H) Each protein level was densitometrically quantified (normalized to 0 min) and shown in the graph below.

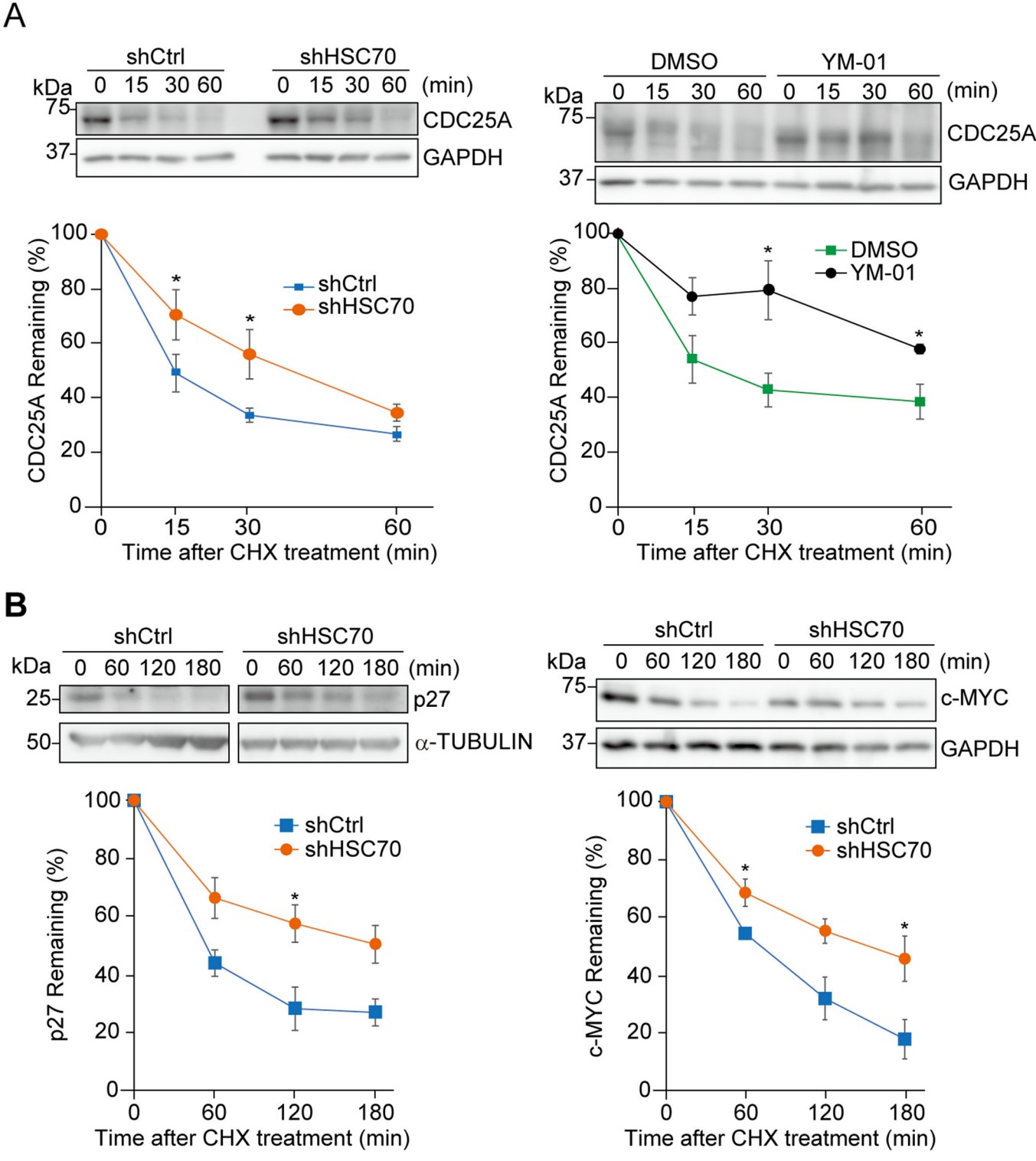

**Figure EV4.  Pharmacological or genetic inhibition of HSC70 prolonged the half-life of other SCF substrates.**

(**A**) CHX chase assay of endogenous CDC25A with or without *HSC70* knockdown, or pharmacological inhibition with YM-01. The bottom graphs show the mean ± s.e.m. *$P < 0.05$ ($n = 3$ independent experiments; Welch's $t$ test). Compared with shCtrl cells or DMSO treatment, a significant difference was observed at 15 min ($P = 0.025$) and 30 min ($P = 0.045$) in shHSC70 cells and at 30 min ($P = 0.042$) and 60 min ($P = 0.011$) in YM-01 treated cells. (**B**) CHX chase assay of endogenous p27 or c-MYC with or without *HSC70* knockdown. The bottom graphs show the mean ± s.e.m. *$P < 0.05$ ($n = 3$ independent experiments, respectively; Welch's $t$ test). For p27 protein, a significant difference was observed at 120 min ($P = 0.032$). For c-MYC protein, significant difference was observed at 60 min ($P = 0.033$) and 180 min ($P = 0.045$). (**A, B**) Each protein level was densitometrically quantified (normalized to 0 min) and shown in the graph below.

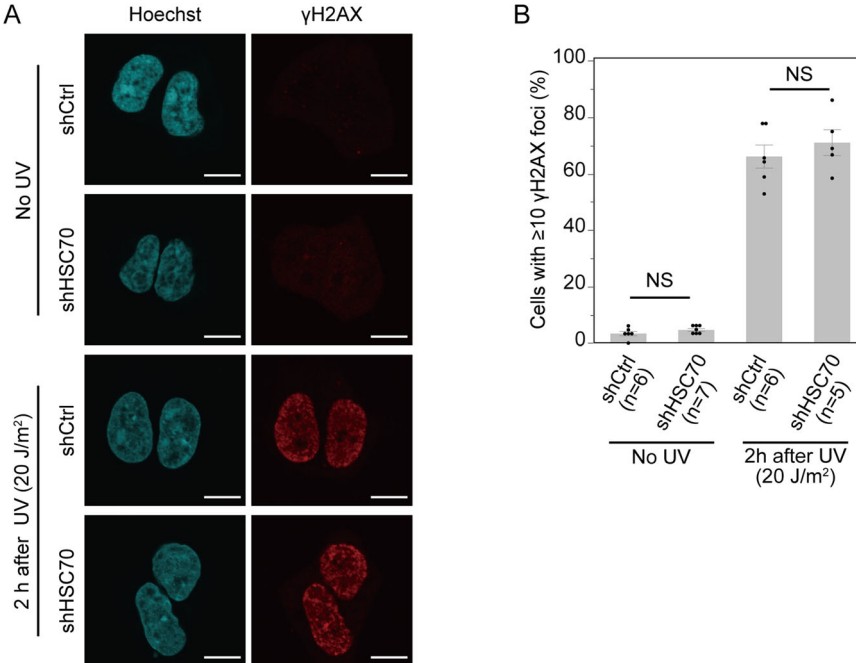

**Figure EV5. Ultraviolet-induced DNA damage in HSC70 knockdown cells.**

(A) Immunostaining of control or HSC70 knockdown HEK293T cells with or without low-dose UV irradiation (20 J/m$^2$) using γH2AX antibody and Hoechst 33342. Scale bars, 10 μm. (B) Quantification of cells with γH2AX positive foci with or without UV irradiation ($n = 6$ independent experiments for shCtrl cells with or without UV, respectively; $n = 7$ or 5 independent experiments for shHSC70 cells with or without UV, respectively; Welch's $t$ test). The graphs show the mean ± s.e.m. NS not significant.

