## [Peer Review File · EMBO Reports]

HSC70 coordinates COP9 signalosome and SCF ubiquitin ligase activity to enable a prompt stress response

Shunsuke Nishimura, Hidetaka Kioka, Shan Ding, Hideyuki Hakui, Haruki Shinomiya, Kazuya Tanabe, Tatsuhiro Hitsumoto, Ken Matsuoka, Hisakazu Kato, Osamu Tsukamoto, Yoshihiro Asano, Seiji Takashima, Radoslav Enchev, and Yasushi Sakata

Corresponding author: Hidetaka Kioka (kioka@cardiology.med.osaka-u.ac.jp)

Review Timeline:

Submission Date:	6th Jun 24
Editorial Decision:	13th Aug 24
Appeal Received:	16th Aug 24
Editorial Decision:	12th Sep 24
Revision Received:	17th Oct 24
Editorial Decision:	3rd Dec 24
Revision Received:	11th Dec 24
Accepted:	23rd Dec 24

Transaction Report:

Dear Dr. Kioka

Thank you for the submission of your research manuscript to EMBO reports. I apologize for the delay in the review process. I have asked 3 referees to assess your manuscript but unfortunately, referee #3 has not yet submitted his/her report. In order to save you from further loss of time, I have decided to make a decision based on the 2 reports we have on your manuscript (see below).

Referee 1 is an expert in the ubiquitin, stress signaling and chaperone field. The referee questions the functional significance and conclusiveness of the findings, given the pleiotropic nature of chaperones and the rather small effect on Hsc70-dependent cullin neddylation. The referee considers further data on the specificity of Hsc70 in CSN and SCF activation by the identification of co-chaperones as crucial to substantiate the conclusions. I have discussed this concern further with referee 1, who emphasized again the necessity to identify co-chaperones to strengthen the conclusiveness of your findings, also given that both, HSP70/90 and CRL ligases regulate so many biological processes. Referee 2 asks to further substantiate the proposed Hsc70 substrate switch and relevance/connection of the Hsc70-CSN and Hsc70-SCF interaction as the current data are not convincing.

Given these concerns, the amount of work required to address them, the uncertain outcome of these experiments, and the fact that EMBO reports can only invite revision of papers that receive enthusiastic support from a majority of referees, I am sorry to say that we cannot offer to publish your manuscript at this point.

I am sorry to disappoint you on this occasion, and hope that the referee comments will be helpful in your continued work in this area.

Yours sincerely

Referee #1:

The manuscript by Nishimura and colleagues describes a role of the Hsc70 chaperone in the regulation of the COP9 signalosome and SCF E3-ligase activity. By using mainly biochemical approaches and experiments in tissue culture the authors show that Hsc70 interacts with the COP9 signalosome enhancing its deneddylation activity towards cullins, and on the other hand with neddylated SCF complex to promote substrate ubiquitination. This function of Hsc70 is specific, Hsp70 independent and also relevant upon stress conditions, namely UV-induced DNA damage. Collectively, the study indicates an alternate role for Hsc70 in CSN and SCF activation.

Overall, this is an intriguing and potentially interesting study. The data are of high quality and in general convincing.

However, I fear the detailed presented model for Hsc70 function, while possible, it is rather speculative, as based on the used approaches the presented effects of Hsc70 may be simply correlative and not inter-linked as currently presented.

The main issue in chaperone research is that studying the effect of chaperone inhibition is not very informative. This is because of the pleiotropic role of chaperones and because inhibition of one chaperone affects the activity of others. The identification of the co-chaperone involved and the generation of point mutants that would discriminate the 2 presented functions of Hsc70 are crucial to define specificity in a chaperone-regulated process.

Currently, it is impossible to exclude that the presented in vitro and in vivo effects of Hsc70 are not due to the fact that Hsc70 directly interferes with substrate ubiquitination by direct or indirect substrate binding. Also, the fact that Hsp70 inhibition does not affect CSN/SCF activity is informative but it is not clear if Hsp70 is involved when Hsc70 is inhibited, as all experiments were performed upon single chaperone inhibition conditions. But even if the authors study whether the effects of Hsc70 inhibition depend on Hsp70, it would still remain uninformative due to the pleiotropic role of chaperones. As mentioned, the identification of co-chaperones and Hsc70 point mutants is a crucial step.

It is also not clear of the additional benefit/role of the Hsc70 in CSN/SCF regulation based on our detailed knowledge on CAND1 and the nedd8 cycle.

In general, the Hsc70-dependent increase in cullin neddylation is rather limited. Studies on the nedd8 pathway inhibitor

MLN4924 showed that 80% change in cullin neddylation is required to provide a biologically significant effect. Therefore, the significance of the Hsc70 effect on SCF neddylation to the observed changes in substrate levels in vivo is not clear.

Fig. 4D: How would Hsc70 increase SCF neddylation in the performed in vitro experiments (Fig. 4)?

Fig. 7: UV causes the phosphorylation of CSN subunits by ATM. This is not taken into consideration on how COP9 signalosome activity may be controlled upon UV. Indeed, the cul1 profile upon UV resembles that upon CSN inhibition (Fig. 3A).

Is the effect of Hsc70 specific for UV? How about other stress conditions that regulate CSN/SCF activity, such as starvation, heat shock etc?

Referee #2:

In the paper entitled "HSC70 alternatively regulates COP9 signalosome and SCF ubiquitin ligase for a prompt stress response", the authors discovered a regulatory function of HSC70 in the COP9 signalosome (CSN)-SCF ubiquitin ligase system, and the stress response. They found HSC70 interacts through its SBD's PBD motif with CSN under basal conditions to promote CSN deneddylation activity. Alternatively, under SCF-activated conditions, HSC70 interacts through its NBD domain with substrate-loaded neddylated SCF to enhance the ubiquitination activity. The binding of HSC70 to CSN or neddylated SCF is regulated by the presence or absence of SCF substrates. In my opinion, this work is interesting, novel, and potentially important for the regulation of SCF function. It did not go further to investigate how the interactions with HSC70 enhance CSN activity or SCF activity, otherwise it could have made more significant impact. Still, this work paved the road for further studies in this direction. The data are of high quality in general; and the paper is well-written and logically presented. I have a few concerns about some of the conclusions of the study.

Major concern:

1. An important point of this study is that available substrate leads to the switch of HSC70's binding target from CSN to neddylated SCF. This point is not fully supported. The data is clear that HSC70 does not bind to non-neddylated SCF or to substrate alone, and that substrate-loaded neddylated SCF reduce HSC70-CSN interaction. However, it remains possible that neddylated SCF complex, even without a loaded substrate, can reduce HSC70-CSN interaction and attract HSC70 to itself. Figure EV5 showing "substrate-free Nedd8-SCF does not inhibit the HSC70-CSN interaction" is unconvincing. In that experiment, the His-CSN pulldown appeared to have pulled down both HSC70 in the presence of substrate-unbound Nedd8-SCF, or did not pulldown CSN itself nor HSC70 without SCF. It seemed the experiment did not work. This figure is presented as supplement, but the conclusion was drawn based on this result.

To prove the point, authors could follow the procedure of Fig5A, but with a no-substrate sample to directly compare substrate-bound (Nedd8-SCF-p-p27) and substrate-free (Nedd8-SCF7) Nedd8-SCF in a HSC70-pulldown/co-IP experiment.

2. What is the connection of the two interactions by HSC70? Is HSC70-CSN interaction a prerequisite for HSC70-SCF interaction, or these are two independent events? If the authors make a point-mutation in the PBD motif of HSC70 to disrupt its interaction with CSN, would the point mutation affect HSC70-NEDD8-SCF interaction? Alternatively, using NR peptide which disrupt HSC70-SCF interaction may also help answering this question.

Minor

3. It appears that the co-immunoprecipitated CUL1 moved slower than neddylated CUL1 on the gel (Fig.3A, Fig.5C...). Can the authors explain that? Are there additional modifications on CUL1?

4. The model shown in figure 7G can be improved to better capture the findings of the study and make it clearer. For examples, are the three SCF drawings in the figure the same or different? Why are there two CSNs and three SCFs? Do they represent their respective abundance?

** As a service to authors, EMBO Press provides authors with the ability to transfer a manuscript that one journal cannot offer to publish to another journal, without the author having to upload the manuscript data again. To transfer your manuscript to another EMBO Press journal using this service, please click on Link Not Available

Dear Dr. Martina Rembold,

I appreciate your and reviewers' effort on reviewing our paper entitled "*HSC70 alternatively regulates COP9 signalosome and SCF ubiquitin ligase for a prompt stress response*" (Manuscript ID; EMBOR-2024-59730V1).

I have carefully reviewed the comments provided and would like to respectfully request a reconsideration of our manuscript based on the following points:

Referee 1 raised a concern regarding the identification of a co-chaperone.

[Redacted]

Moreover, we have generated a loop mutant of HSC70, which is almost equivalent to the point mutant suggested by referee 1. This HSC70 loop mutant cannot bind to NR peptide, HSC70 model substrate (PMID: 22157767).

As anticipated, this loop mutant fails to interact with CSN and does not activate the neddylation activity of CSN.

We are currently preparing to test whether this loop mutant can bind to and activate NEDD-SCF ubiquitination activity.

In light of these new data, I kindly request you to discuss with referee 1 again to whether these findings address

his/her major concerns, and hopefully, consider inviting us to submit a revised manuscript with point-by-point response.

Given the positive comments from referees 1 and 2 in their summarizing paragraph, we believe that our paper is a strong candidate for publication in EMBO Reports.

Thank you once again for your time and consideration.

I would greatly appreciate your feedback at your earliest convenience.

Best regards,
Hidetaka Kioka

Hidetaka Kioka M.D., Ph.D.
Assistant Professor
Department of Cardiovascular Medicine
Osaka University Graduate School of Medicine
2-2 Yamadaoka, Suita Osaka, 565-0871 JAPAN
TEL: [+81-6-6879-3640](tel:+81-6-6879-3640), FAX: [+81-6-6879-3639](tel:+81-6-6879-3639)
E-mail:
kioka@cardiology.med.osaka-u.ac.jp
kioka.hidetaka.med@osaka-u.ac.jp

Figures R1, R2 and R3 for reviewers were removed.

Preliminary responses to the referee's comment

We deeply appreciate referee's insightful comments especially focusing on co-chaperone and Hsc70 point mutant. According to referee's comments, we have already performed several experiments to respond to the major concerns, and the results are detailed in this preliminary response. If this preliminary response is enough to invite revision, we will be happy to submit final version of point-by-point response along with a revised manuscript.

[Comment]

The manuscript by Nishimura and colleagues describes a role of the Hsc70 chaperone in the regulation of the COP9 signalosome and SCF E3-ligase activity. By using mainly biochemical approaches and experiments in tissue culture the authors show that Hsc70 interacts with the COP9 signalosome enhancing its deneddylating activity towards cullins, and on the other hand with neddylated SCF complex to promote substrate ubiquitination. This function of Hsc70 is specific, Hsp70 independent and also relevant upon stress conditions, namely UV-induced DNA damage. Collectively, the study indicates an alternate role for Hsc70 in CSN and SCF activation.

Overall, this is an intriguing and potentially interesting study. The data are of high quality and in general convincing.

[Response]

We appreciate your overall favorable evaluation, and the constructive comments provided. Your encouraging words motivate us to continue our work.

[Comment]

However, I fear the detailed presented model for Hsc70 function, while possible, it is rather speculative, as based on the used approaches the presented effects of Hsc70 may be simply correlative and not inter-linked as currently presented.

[Response]

Thank you for highlighting this important issue. To determine whether the effects of HSC70 on CSN and NEDD8-SCF are interlinked, we will investigate if the baseline interaction between HSC70 and CSN is necessary for the shift of HSC70 to NEDD8-SCF under CSN5i-3 treatment, using an HSC70 mutant that cannot bind to CSN. This investigation is expected to strengthen our model for HSC70 function. We will include these findings in the final version of point-by-point response.

[Comment]

The main issue in chaperone research is that studying the effect of chaperone inhibition is not very informative. This is because of the pleiotropic role of chaperones and because inhibition of one chaperone affects the activity of others. The identification of the co-chaperone involved and the generation of point mutants that would discriminate the 2 presented functions of Hsc70 are crucial to define specificity in a chaperone-regulated process.

[Response]

Thank you for your insightful comments. Following your suggestion, we aimed to specify the two functions of HSC70 by identifying a co-chaperone and utilizing an HSC70 point mutant.

Figure R1 and associated information removed

Figure R2 removed

Figure R3 removed

Next, to further clarify whether HSC70 activates CSN in an ATPase-dependent or independent manner, we used ATPase-dead point mutant of HSC70 (HSC70 E175S). While HSC70 E175S cannot interact with (Fig. 4D) or activate SCF (Fig. 4E), it can interact with CSN (Fig. R4) and activate the de-neddylolation activity of CSN (Fig. R5). These findings suggest that HSC70 activates CSN in an ATPase-independent manner.

Fig. R4

In vitro pulldown assay.

A mixture containing Strep-tagged CSN and GST-HSC70 WT or E175S was subjected to immunoprecipitation with Streptactin Sepharose.

Fig. R5

In vitro deneddylation assay.

CSN deneddylation activity with or without HSC70 E175S. Neddylated or non-neddylated CUL1 were detected by immunoblotting with anti-CUL1 antibody.

Information for reviewers redacted

[Comment]

Currently, it is impossible to exclude that the presented in vitro and in vivo effects of Hsc70 are not due to the fact that Hsc70 directly interferes with substrate ubiquitination by direct or indirect substrate binding. Also, the fact that Hsp70 inhibition does not affect CSN/SCF activity is informative but it is not clear if Hsp70 is involved when Hsc70 is inhibited, as all experiments were performed upon single chaperone inhibition conditions. But even if the authors study whether the effects of Hsc70 inhibition depend on Hsp70, it would still remain uninformative due to the pleiotropic role of chaperones. As mentioned, the identification of co-chaperones and Hsc70 point mutants is a crucial step.

[Response]

Thank you for your insightful comments. We agree with you that the identifying of co-chaperones and HSC70 point mutants is crucial to strengthen our model. By incorporating in vitro experiments using co-chaperones and HSC70 point mutant, we believe that we can adequately address the referee's concerns.

[Comment]

It is also not clear of the additional benefit/role of the Hsc70 in CSN/SCF regulation based on our detailed knowledge on CAND1 and the nedd8 cycle.

[Response]

Thank you for your insightful comments. As you pointed out, CAND1-mediated disassembly/assembly and NEDD8 cycle are important for activating the SCF cycle. We showed HSC70 cannot interact with Non-neddylated SCF (Fig. 3). Therefore, if HSC70 is involved in CAND1-mediated SCF disassembly/assembly, HSC70 should interact with CAND1. We will examine this possibility by immunoprecipitation assay in the final version of point-of-point response.

[Comment]

In general, the Hsc70-dependent increase in cullin neddylation is rather limited. Studies on the nedd8 pathway inhibitor MLN4924 showed that 80% change in cullin neddylation is required to provide a biologically significant effect. Therefore, the significance of the Hsc70 effect on SCF neddylation to the observed changes in substrate levels in vivo is not clear.

[Response]

Thank you for your comment. We agree that the significance of HSC70's effect on CSN and SCF in vivo should be further discussed. As you pointed, the effect of HSC70 on cullin

neddylation seems limited, raising concerns about biological significance. Indeed, without UV stress, we did not observe a significant difference in cell proliferation between control and HSC70 KD cells (Fig. 7D. left panel), despite the elongated half-life of SCF substrate (Fig. 7B). However, UV stress revealed the significance of HSC70 in vivo, as shown in Fig. 7C, 7D (right panel), 7E, and 7F.

As we discussed at the 1st paragraph in the discussion section, the alternative interaction between HSC70 and CSN or SCF can be beneficial for cells for the following two reasons:

1. Cells can prepare to the upcoming stress at baseline through HSC70-CSN interaction.
2. Cells can promptly respond to UV stress via HSC70-mediated SCF activation.

This explains why the biological significance of the HSC70-CSN-CSF interplay becomes evident only under stress conditions. Specifically, the delayed degradation of CDC25a in HSC70 knockdown cells was notably more pronounced under UV stress (Fig. 7B) compared to the cycloheximide (CHX) chase assay without UV exposure (Fig. EV7A).

Although we cannot completely exclude the involvement of the known functions of HSC70 or other chaperones, we believe that this is a plausible explanation, supported by both in vitro and in vivo data. We would appreciate your understanding.

[Comment]

Fig. 4D: How would Hsc70 increase SCF neddylation in the performed in vitro experiments (Fig. 4)?

[Response]

Thank you for your comment. I apologize for any misunderstanding, but I am having difficulty understanding your specific point. We performed this in vitro pulldown assay to demonstrate the interaction between HSC70 E175S and NEDD8-SCF (with WT as the positive control and Δ NBD (Δ 1-386 aa) as the negative control), rather than to show an increase in SCF neddylation. Could you please clarify or rephrase your comment?

[Comment]

Fig. 7: UV causes the phosphorylation of CSN subunits by ATM. This is not taken into consideration on how COP9 signalosome activity may be controlled upon UV. Indeed, the cul1 profile upon UV resembles that upon CSN inhibition (Fig. 3A).

[Response]

Thank you for your insightful comments. We agree with you that the phosphorylation of CSN by UV should be taken into consideration. As shown in Fig. 7A, UV stress dissociates the interaction between HSC70 and CSN (Fig. 7A, middle panel). On the contrary, incubated with MLN4924, UV stress did not affect the interaction between HSC70 and CSN (Fig. 7A right

panel). Furthermore, the shift of HSC70 from CSN to NEDD8-SCF occurs even under CSN5i-3 treatment (Fig. 3A) and forced expression of substrates (Fig. 5B, C), both of which would not cause CSN phosphorylation. These findings suggest that UV induced phosphorylation of CSN should not be essential in the molecular interplay among HSC70, CSN, and NEDD8-SCF.

[Comment]

Is the effect of Hsc70 specific for UV? How about other stress conditions that regulate CSN/SCF activity, such as starvation, heat shock etc?

[Response]

Based on your comment, we will try several stress conditions and examine whether the effect of HSC70 is specific for UV or applicable for other types of stress in the final version of point-by-point response.

Dear Dr. Kioka

Thank you for your mail asking us to consider a revision of your manuscript. I have meanwhile received feedback on your proposed revision plan from Referee #1 who is overall positive that his/her key concerns, including the definition of co-chaperones and dissecting the activities of Hsc70 in SCF activity, could be addressed by the proposed plan.

Given this positive feedback, I would like to give you the chance to revise your study for EMBO Reports along the lines you suggested.

Acceptance of the manuscript will depend on a positive outcome of a second round of review. It is EMBO reports policy to allow a single round of major revision only and acceptance or rejection of the manuscript will therefore depend on the completeness of your responses included in the next, final version of the manuscript.

In the interest of protecting the conceptual advance provided by the work, we recommend a revision within 3 months (12th Dec 2024). Please discuss the revision progress ahead of this time with the editor if you require more time to complete the revisions.

Please find below detailed instructions on the formatting of the revised manuscript.

*****IMPORTANT NOTE:

We perform an initial quality control of all revised manuscripts before re-review. Your manuscript will FAIL this control and the handling will be delayed IN CASE the following APPLIES:

- 1) A data availability section providing access to data deposited in public databases is missing. If you have not deposited any data, please add a sentence to the data availability section that explains that.
- 2) Your manuscript contains statistics and error bars based on $n=2$. Please use scatter blots in these cases. No statistics should be calculated if $n=2$.

When submitting your revised manuscript, please carefully review the instructions that follow below. Failure to include requested items will delay the evaluation of your revision.*****

2) individual production quality figure files as .eps, .tif, .jpg (one file per figure).

Please download our Figure Preparation Guidelines (figure preparation pdf) from our Author Guidelines pages <https://www.embopress.org/page/journal/14693178/authorguide> for more info on how to prepare your figures.

4) a complete author checklist, which you can download from our author guidelines (<<https://www.embopress.org/page/journal/14693178/authorguide>>). Please insert information in the checklist that is also reflected in the manuscript. The completed author checklist will also be part of the RPF.

5) Please note that all corresponding authors are required to supply an ORCID ID for their name upon submission of a revised manuscript (<<https://orcid.org/>>). Please find instructions on how to link your ORCID ID to your account in our manuscript tracking system in our Author guidelines (<<https://www.embopress.org/page/journal/14693178/authorguide#authorshipguidelines>>)

6) We replaced Supplementary Information with Expanded View (EV) Figures and Tables that are collapsible/expandable online. A maximum of 5 EV Figures can be typeset. EV Figures should be cited as 'Figure EV1, Figure EV2' etc... in the text and their respective legends should be included in the main text after the legends of regular figures.

7) Please include a dedicated "Data Availability" section at the end of the Methods (suggested wording: "The [structural coordinates | microarray | mass spectrometry] data from this publication have been deposited to the [name of the database] database [URL] and assigned the identifier [accession | permalink | hashtag]."). Should this not apply, this should still be stated as "This study includes no data deposited in external repositories."

Additional information on source data and instruction on how to label the files are available
<<https://www.embopress.org/page/journal/14693178/authorguide#sourcedata>>.

10) Figure legends and data quantification:
The following points must be specified in each figure legend:

- the name of the statistical test used to generate error bars and P values,
 - the number (n) of independent experiments (please specify technical or biological replicates) underlying each data point,
 - the nature of the bars and error bars (s.d., s.e.m.)
-
- If the data are obtained from n {less than or equal to} 5, show the individual data points in addition to the SD or SEM.
 - If the data are obtained from n {less than or equal to} 2, use scatter blots showing the individual data points.

See also the guidelines for figure legend preparation:
<https://www.embopress.org/page/journal/14693178/authorguide#figureformat>

11) Our journal encourages inclusion of *data citations in the reference list* to directly cite datasets that were re-used and obtained from public databases. Data citations in the article text are distinct from normal bibliographical citations and should directly link to the database records from which the data can be accessed. In the main text, data citations are formatted as follows: "Data ref: Smith et al, 2001" or "Data ref: NCBI Sequence Read Archive PRJNA342805, 2017". In the Reference list, data citations must be labeled with "[DATASET]". A data reference must provide the database name, accession number/identifiers and a resolvable link to the landing page from which the data can be accessed at the end of the reference. Further instructions are available at <<https://www.embopress.org/page/journal/14693178/authorguide#referencesformat>>.

12) All Materials and Methods need to be described in the main text using our 'Structured Methods' format. According to this format, the Methods section includes a Reagents and Tools Table (listing key reagents, experimental models, software and relevant equipment and including their sources and relevant identifiers) followed by a Methods and Protocols section describing the methods, ideally using a step-by-step protocol format. The aim is to facilitate adoption of the methodologies across labs. Please download and fill our Reagents and Tools Table template (.docx), which you can find in our author guidelines: <https://www.embopress.org/page/journal/14693178/authorguide#structuredmethods>. When submitting your revised manuscript, please do not include the Reagents and Tools Table in the Methods section of the manuscript but upload it as a separate file choosing the file type "Reagent Table". An example of a Method paper with Structured Methods can be found here: <https://www.embopress.org/doi/10.15252/msb.20178071>.

13) As part of the EMBO publication's Transparent Editorial Process, EMBO Reports publishes online a Review Process File to accompany accepted manuscripts. This File will be published in conjunction with your paper and will include the referee reports, your point-by-point response and all pertinent correspondence relating to the manuscript.

You are able to opt out of this by letting the editorial office know (emboreports@embo.org). If you do opt out, the Review Process File link will point to the following statement: "No Review Process File is available with this article, as the authors have

chosen not to make the review process public in this case."

Kind regards,

Responses to the referees' comment

We thank the reviewers for the positive and constructive comments about our manuscript, "HSC70 alternatively regulates COP9 signalosome and SCF ubiquitin ligase for a prompt stress response (Manuscript ID; EMBOR-2024-59730V2-Q)", by Nishimura et al. We performed additional experiments and revised the manuscript in response to the comments. We hope that we have responded adequately and that the revised manuscript is now suitable for publication.

Response to referee 1

[Comment]

The manuscript by Nishimura and colleagues describes a role of the Hsc70 chaperone in the regulation of the COP9 signalosome and SCF E3-ligase activity. By using mainly biochemical approaches and experiments in tissue culture the authors show that Hsc70 interacts with the COP9 signalosome enhancing its deneddylating activity towards cullins, and on the other hand with neddylated SCF complex to promote substrate ubiquitination. This function of Hsc70 is specific, Hsp70 independent and also relevant upon stress conditions, namely UV-induced DNA damage. Collectively, the study indicates an alternate role for Hsc70 in CSN and SCF activation.

Overall, this is an intriguing and potentially interesting study. The data are of high quality and in general convincing.

[Response]

We appreciate your overall favorable evaluation, and the constructive comments provided. Your encouraging words motivate us to continue our work.

[Comment]

However, I fear the detailed presented model for Hsc70 function, while possible, it is rather speculative, as based on the used approaches the presented effects of Hsc70 may be simply correlative and not inter-linked as currently presented.

[Response]

Thank you for highlighting this important issue. To determine whether the effects of HSC70 on CSN and NEDD8-SCF are interlinked, we have investigated if the baseline interaction between HSC70 and CSN is necessary for the shift of HSC70 to NEDD8-SCF under CSN5i-3 treatment, using an HSC70 Δ PBD mutant that cannot bind to CSN.

To test this, we performed immunoprecipitation using HEK293T cells transiently

transfected with FLAG-tagged HSC70 wild type (WT) or the Δ PBD mutant (New Fig. 5A). Under control conditions, FLAG-HSC70 WT interacts with CSN, whereas the Δ PBD mutant (Δ 405-437aa) does not. Upon SCF activation via CSN5i-3 treatment, FLAG-HSC70 WT shifts its interaction to NEDD8-SCF. In contrast, the Δ PBD mutant fails to bind NEDD8-SCF. Considering that the Δ PBD mutant binds to NEDD8-SCF in vitro (Fig. 3C), convergence of HSC70 to NEDD8-SCF by CSN would be necessary for HSC70 to interact with NEDD8-SCF in vivo. These findings suggest that the interaction between HSC70 and CSN under basal conditions is critical for enabling HSC70 to interact with NEDD8-SCF. Thanks to your insightful comments, we successfully demonstrated that the two functions of HSC70 are interconnected rather than independent. These results are included in the text and Fig. 5A in the revised manuscript.

A

New Fig.5A Co-IP using anti-FLAG M2 agarose in HEK293T cells transiently transfected with the indicated HSC70 mutants. If indicated, pre-treatment with 500 nM CSN5i-3 for 10 min was performed.

[Comment]

The main issue in chaperone research is that studying the effect of chaperone inhibition is not very informative. This is because of the pleiotropic role of chaperones and because inhibition of one chaperone affects the activity of others. The identification of the co-chaperone involved and the generation of point mutants that would discriminate the 2 presented functions of Hsc70 are crucial to define specificity in a chaperone-regulated process.

[Response]

Thank you for your insightful comments. Following your suggestion, we aimed to specify the two functions of HSC70 by 1) identifying a co-chaperone and 2) utilizing an HSC70 point mutant.

Figure for reviewers and associated information removed

Figures for reviewers removed

- 2) The ATPase-dead point mutant of HSC70 activates CSN, whereas it cannot activate SCF.

To further differentiate the two functions of HSC70, we used the ATPase-dead point mutant of HSC70 (HSC70 E175S) in the in vitro de-neddylation assay. While
HSC70

E175S cannot interact with (Fig. 4D) or activate SCF (Fig. 4E), it can interact with CSN (New Fig. 2C) and activate the de-neddylation activity of CSN (New Fig. 2D). These findings suggest that HSC70 activates CSN in an ATPase-independent manner, contrasting its ATPase-dependent role in SCF ubiquitination activity. These results are included in the text and Fig. 2C and 2D in the revised manuscript.

C

New Fig. 2C

In vitro pulldown assay. A mixture containing Strep-tagged CSN and GST-HSC70 WT or E175S was subjected to immunoprecipitation with Streptactin Sepharose.

D

New Fig. 2D

In vitro deneddylation assay. CSN deneddylation activity with or without HSC70 E175S. Neddylated or non-neddylated CUL1 were detected by immunoblotting with anti-CUL1 antibody.

Information for reviewers redacted

[Comment]

Currently, it is impossible to exclude that the presented in vitro and in vivo effects of Hsc70 are not due to the fact that Hsc70 directly interferes with substrate ubiquitination by direct or indirect substrate binding. Also, the fact that Hsp70 inhibition does not affect CSN/SCF activity is informative but it is not clear if Hsp70 is involved when Hsc70 is inhibited, as all experiments were performed upon single chaperone inhibition conditions. But even if the authors study whether the effects of Hsc70 inhibition depend on Hsp70, it would still remain uninformative due to the pleiotropic role of chaperones. As mentioned, the identification of co-chaperones and Hsc70 point mutants is a crucial step.

[Response]

Thank you for your insightful comments. We agree with you that the identifying of co-chaperones and HSC70 point mutants can strengthen our model. By adding in vitro experiments using HSC70 point mutant into the revised manuscript, we believe that we can adequately address the referee's concerns.

[Comment]

It is also not clear of the additional benefit/role of the Hsc70 in CSN/SCF regulation based on our detailed knowledge on CAND1 and the nedd8 cycle.

[Response]

Thank you for your insightful comments. As you pointed out, CAND1-mediated disassembly/assembly and the NEDD8 cycle are important for activating the SCF cycle. We showed HSC70 cannot interact with non-neddylated SCF (Fig. 3). Therefore, if HSC70 is involved in CAND1-mediated SCF disassembly/assembly, HSC70 should interact with CAND1. To test this, we performed immunoprecipitation using an anti-HSC70 antibody, and no interaction between HSC70 and CAND1 was observed (Reviewer-only Fig. 4). These data suggest that HSC70 does not influence CAND1-mediated disassembly.

Reviewer-only Fig. 4 Co-IP in HEK293T cells using HSC70 antibody, with rabbit-IgG serving as a negative control.

[Comment]

In general, the Hsc70-dependent increase in cullin neddylation is rather limited. Studies on the nedd8 pathway inhibitor MLN4924 showed that 80% change in cullin neddylation is required to provide a biologically significant effect. Therefore, the significance of the Hsc70 effect on SCF neddylation to the observed changes in substrate levels in vivo is not clear.

[Response]

Thank you for your comment. As you pointed, the effect of HSC70 on cullin neddylation seems limited, raising concerns about biological significance. Indeed, without UV stress, we did not observe a significant difference in cell proliferation between control and HSC70 KD cells (Fig. 7D. left panel), despite the elongated half-life of SCF substrate (Fig. 7B). However, UV stress revealed the significance of HSC70 in vivo, as shown in Fig. 7C, 7D (right panel), 7E, and 7F.

As we discussed at the first paragraph in the discussion section, the alternative interaction between HSC70 and CSN or SCF can be beneficial for cells for the following two reasons:

1. Cells can prepare to the upcoming stress at baseline through HSC70-CSN interaction.
2. Cells can promptly respond to UV stress via HSC70-mediated SCF activation.

This explains why the biological significance of the HSC70-CSN-CSF interplay becomes evident only under stressed conditions. Specifically, the delayed degradation of CDC25a in HSC70 knockdown cells was notably more pronounced under UV stress (Fig. 7B) compared to the cycloheximide (CHX) chase assay without UV exposure (Fig. EV7A).

Although we cannot completely exclude the involvement of the known functions of HSC70 or other chaperones, we believe that this is a plausible explanation, supported by both in vitro and in vivo data. We would appreciate your understanding.

[Comment]

Fig. 4D: How would Hsc70 increase SCF neddylation in the performed in vitro experiments (Fig. 4)?

[Response]

Thank you for your comment. We apologize for any misunderstanding. We performed this in vitro pulldown assay to demonstrate the interaction between HSC70 E175S and NEDD8-SCF (with WT as the positive control and Δ NBD (Δ 1-386 aa) as the negative control), rather than to show an increase in SCF neddylation. We would appreciate your understanding.

[Comment]

Fig. 7: UV causes the phosphorylation of CSN subunits by ATM. This is not taken into consideration on how COP9 signalosome activity may be controlled upon UV. Indeed, the cul1 profile upon UV resembles that upon CSN inhibition (Fig. 3A).

[Response]

Thank you for your insightful comments. We agree with you that the phosphorylation of CSN by UV should be taken into consideration. As shown in Fig. 7A, UV stress dissociates the interaction between HSC70 and CSN (Fig. 7A, middle panel). On the contrary, incubated with MLN4924, UV stress did not affect the interaction between HSC70 and CSN (Fig. 7A right panel). Furthermore, the shift of HSC70 from CSN to NEDD8-SCF occurs even under CSN5i-3 treatment (Fig. 3A) and forced expression of substrates (Fig. 5B, C), both of which would not cause CSN phosphorylation. These findings suggest that UV induced phosphorylation of CSN should not be essential in the molecular interplay among HSC70, CSN, and NEDD8-SCF.

[Comment]

Is the effect of Hsc70 specific for UV? How about other stress conditions that regulate CSN/SCF activity, such as starvation, heat shock etc?

[Response]

Based on your comment, we investigated whether nutrient starvation stress induces a shift of HSC70 from CSN to NEDD8-SCF. After one hour of starvation in Hank's balanced salt solution (HBSS), we observed a decrease in the interaction between HSC70 and CSN, coupled with the emergence of interaction between HSC70 and NEDD8-CUL1 (Reviewer-only Fig. 5). Similarly, doxorubicin (DOX) treatment, which increase oxidative stress and DNA damage, also reduced the interaction between HSC70 and CSN, while increasing the interaction with NEDD8-CUL1. These findings are consistent with the effects observed

under UV stress, indicating that the role of HSC70 in CSN-SCF regulation is not specific to UV stress. Therefore, we are confident that our conclusions remain valid regardless of the type of stress that activates SCF. We will thoroughly investigate the detailed functional role of HSC70 under various SCF-activating conditions in different cells in future studies.

Reviewer-only Fig. 5

Co-IP in HEK293T cells using HSC70 antibody, with IgG serving as a negative control. HEK293T cells were starved with HBSS for one hour or treated with 10 μ M DOX for one hour.

Response to referee 2

[Comment]

In the paper entitled "HSC70 alternatively regulates COP9 signalosome and SCF ubiquitin ligase for a prompt stress response", the authors discovered a regulatory function of HSC70 in the COP9 signalosome (CSN)-SCF ubiquitin ligase system, and the stress response. They found HSC70 interacts through its SBD's PBD motif with CSN under basal conditions to promote CSN deneddylation activity. Alternatively, under SCF-activated conditions, HSC70 interacts through its NBD domain with substrate-loaded neddylated SCF to enhance the ubiquitination activity. The binding of HSC70 to CSN or neddylated SCF is regulated by the presence or absence of SCF substrates. In my opinion, this work is interesting, novel, and potentially important for the regulation of SCF function. It did not go further to investigate how the interactions with HSC70 enhance CSN activity or SCF activity, otherwise it could have made more significant impact. Still, this work paved the road for further studies in this direction. The data are of high quality in general; and the paper is well-written and logically presented. I have a few concerns about some of the conclusions of the study.

[Response]

We appreciate your overall favorable evaluation, and the constructive comments provided.

[Comment]

Major concern:

1. An important point of this study is that available substrate leads to the switch of HSC70's binding target from CSN to neddylated SCF. This point is not fully supported. The data is clear that HSC70 does not bind to non-neddylated SCF or to substrate alone, and that substrate-loaded neddylated SCF reduce HSC70-CSN interaction. However, it remains possible that neddylated SCF complex, even without a loaded substrate, can reduce HSC70-CSN interaction and attract HSC70 to itself. Figure EV5 showing "substrate-free Nedd8-SCF does not inhibit the HSC70-CSN interaction" is unconvincing. In that experiment, the His-CSN pulldown appeared to have pulled down both HSC70 in the presence of substrate-unbound Nedd8-SCF, or did not pulldown CSN itself nor HSC70 without SCF. It seemed the experiment did not work. This figure is presented as supplement, but the conclusion was drawn based on this result.

To prove the point, authors could follow the procedure of Fig5A, but with a no-substrate sample to directly compare substrate-bound (Nedd8-SCF-p-p27) and

substrate-free (Nedd8-SCF7) Nedd8-SCF in a HSC70-pulldown/co-IP experiment.

[Response]

Thank you for your insightful comment. Per your suggestion, we performed the in vitro pulldown assay without the substrate (p-p27). Since wild-type CSN deneddylate substrate-free neddylated SCF, we performed GST-pulldown assay in the presence of CSN inhibitor, CSN5i-3. GST-HSC70 pulldown assay with Strep-tagged CSN in the presence of substrate-free NEDD8-SCF revealed that the binding between HSC70 and CSN was not attenuated by substrate-free NEDD8-SCF, but rather HSC70 and CSN both bound to substrate-free NEDD8-SCF (New Fig. 5C). These findings indicated that the presence of substrate determines the interaction among HSC70, CSN, and neddylated SCF. In the presence of substrate, HSC70 switches its interacting protein from CSN to substrate-bound neddylated SCF. In the absence of substrate, HSC70 and CSN can bind to substrate-free neddylated SCF, thereby enhancing the deneddylation activity of CSN. Thanks to your insightful comments, we successfully demonstrated the substrate-dependent interaction between HSC70 and CSN or neddylated SCF. We have added this data as New Fig. 5C in the revised manuscript, replacing Fig. EV5.

C

New Fig. 5C

In vitro pulldown assay. A mixture containing GST-tagged HSC70 mutants, Strep-tagged CSN and NEDD8-SCF (substrate-free) in the presence of 1 μ M CSN5i-3 was subjected to immunoprecipitation with Glutathione Sepharose 4 Fast Flow.

[Comment]

2. What is the connection of the two interactions by HSC70? Is HSC70-CSN interaction a prerequisite for HSC70-SCF interaction, or these are two independent events? If the authors make a point-mutation in the PBD motif of HSC70 to disrupt its interaction with CSN, would the point mutation affect HSC70-NEDD8-SCF interaction? Alternatively, using NR peptide which disrupt HSC70-SCF interaction may also help answering this question.

[Response]

Thank you for your comment. We used Δ PBD mutant of HSC70, which prevents its interaction with CSN under basal condition. We then examined whether this mutant could interact with NEDD-SCF under CSN5i-3 treatment. While HSC70 WT interacts with NEDD-

SCF under CSN5i treatment, the mutant does not (New Fig. 5A). Since HSC70 interacts with NEDD-SCF through the NBD domain, this suggests that the HSC70-CSN interaction at baseline is essential for HSC70 to interact with NEDD-SCF under stressed conditions. Thanks to your insightful comments, we successfully demonstrated that the two functions of HSC70 are interconnected rather than independent. We have added this new data in the revised manuscript.

A

New Fig.5A Co-IP using anti-FLAG M2 agarose in HEK293T cells transiently transfected with the indicated HSC70 mutants. If indicated, pre-treatment with 500 nM CSN5i-3 for 10 min was performed.

[Comment]

Minor

3. It appears that the co-immunoprecipitated CUL1 moved slower than neddylated CUL1 on the gel (Fig.3A, Fig.5C...). Can the authors explain that? Are there additional modifications on CUL1?

[Response]

Thank you for pointing this out. We currently do not have a clear explanation for this observation. It is possible that there are additional modifications on CUL1. We will investigate this issue in future studies. We appreciate your insightful comment and will ensure to address it thoroughly. Thank you again for your valuable feedback.

[Comment]

4. The model shown in figure 7G can be improved to better capture the findings of the study and make it clearer. For examples, are the three SCF drawings in the figure the same or different? Why are there two CSNs and three SCFs? Do they represent their respective abundance?

[Response]

Thank you for your comment. Following your suggestion, we have improved the schema (Fig. 7G) to better capture our findings.

Dear Dr. Kioka

Thank you for the submission of your revised manuscript to EMBO reports. I have already forwarded you the referee reports and you indicated that you can and will address the remaining issues raised by Reviewer #1 by conducting the two suggested experiments. I copy the referee reports again below my signature.

From the editorial side there are also a number things that we need, as follows:

- Please provide up to 5 keywords.
- Regarding the Author Contributions, we now use CRediT to specify the contributions of each author in the journal submission system. Therefore, please remove the Author Contributions from the manuscript file and make sure that the author contributions in our online manuscript tracking system are correct and up-to-date. The information you specified in the system will be automatically retrieved and typeset into the article. You can enter additional information in the free text box provided, if you wish.
- Please provide a complete author checklist, which you can download from our author guidelines (<<https://www.embopress.org/page/journal/14693178/authorguide>>). Please insert information in the checklist that is also reflected in the manuscript. The completed author checklist will also be part of the RPF.
- The funding information needs to be part of the Acknowledgments.
- Please also complete the information on funding in the online manuscript tracking system. Currently, the following information is missing: The Takeda Science Foundation, SENSIN Medical Research Foundation, The Osaka Medical Research Foundation for Intractable Diseases, Japan Heart Foundation Research Grant for Dilated Cardiomyopathy, The UK Medical Research Council.
- We can only typeset up to 5 EV figures. Please either combine some of the EV figures to reduce their number or alternatively, move some of the data into an Appendix (single PDF including figures and their legends with table of content and page numbers on the title page).
- The Data availability statement should only refer to the data that were deposited in public repositories. Therefore, please remove the statements referring to the main text of Source Data Files.
- Please note that the specific URL for PXD042742 dataset needs to be provided in the data availability statement.
- Please upload the source data as one folder per figure.
- The manuscript sections should be in the following order: Title page - Abstract & Keywords - Introduction - Results - Discussion - Methods - Data Availability - Acknowledgments - Disclosure Statement & Competing Interests - References - Figure Legends - (Main Tables with legends if applicable) - Expanded View Figure Legends.
- Supplementary Figure Legends should be updated to Expanded View Figure Legends.
- Our production/data editors have asked you to clarify several points in the figure legends (see below). Please incorporate these changes in the manuscript and return the revised file with tracked changes with your final manuscript submission.

A) Statistical test information. Only p-values that are actually shown in the figure panel(s) should (and must) be defined in the legends, all others should be removed from (or added to) the legend. Moreover, we ask for the specification of exact p-values:

- Please note that the exact p values are not provided in the legends of figures 2a, d; 6a-c; 7b, d-f; EV 6a-b.
- Please note that in figures 2b; 6a; EV 7b; there is a mismatch between the annotated p values in the figure legend and the annotated p values in the figure file that should be corrected.

B) Replicates and error bars:

- Please note that the box plots need to be defined in terms of minima, maxima, centre, bounds of box and whiskers, and percentile in the legend of figure 2e.
- Please note that information related to n is missing in the legends of figures 6a; EV 4b-c, f; EV 7b.
- Although 'n' is provided, please describe the nature of entity for 'n' in the legends of figures 6b-c; EV 4e.

- As a standard procedure, we edit the title and abstract of manuscripts to make them more accessible to a general readership. Please find the edited versions below my signature for your review.

- Finally, EMBO Reports papers are accompanied online by

- A) a short (1-2 sentences) summary of the findings and their significance,
- B) 2-3 bullet points highlighting key results and
- C) a schematic summary figure that provides a sketch of the major findings (not a data image).

Please provide the summary figure as a separate file in PNG or JPG format at a size of 550x300-600 pixels (width x height). Please note that the size is rather small and that text needs to be readable at the final size. Please send us this information along with the revised manuscript.

- On a different note, I would like to alert you that EMBO Press offers a new format for a video-synopsis of work published with us, which essentially is a short, author-generated film explaining the core findings in hand drawings, and, as we believe, can be very useful to increase visibility of the work. This has proven to offer a nice opportunity for exposure i.p. for the first author(s) of the study. Please see the following link for representative examples and their integration into the article web page:

<https://www.embopress.org/doi/full/10.15252/emboj.2019103932>

With kind regards,

=====

Referee #1:

The revised manuscript by Nishimura et al. has been significantly improved with the addition of several new data including the use of HSC70 mutants that discriminate the role of HSC70 in CSN and SCF-Nedd8-substrate function.

The in vitro data are now very convincing. However, in the absence of data regarding potential co-chaperones (the authors provide their reasoning for this) the presented data in cells are still based on looking the effect of HSC70 inhibition on protein stability, which as previously discussed are not very informative. If possible, this part will be strengthened potentially with the expression of HSC70 mutants such as the ATPase-defective mutant (HSC70 E175S) in HSC70 knockdown cells.

Also, the effect of HSC70 knockdown on substrate stability has to be presented in context of the effect of CSN3 inhibitors. The reviewer acknowledges that CSN inhibition could block SCF activity through sustained cullin hyper-neddylation but this has to be experimentally addressed and discussed.

Referee #2:

The revised manuscript has clarified a few gaps and significantly improved the story. With new data, they have nicely addressed the dynamic and consequential interplays between HSC70, CSN, and SCF complexes. I think the study has convincingly delineate a novel and important role of HSC70, member of HSP70 chaperones, in the SCF/CSN-mediated pathway of protein degradation. Both SCF/CSN and HSP chaperones are of fundamental importance to cell functions, and both have pleiotropic phenotypes. I think the study has pointed to an emerging mechanism by which the chaperone manages and coordinates activities of multi-subunit complexes. It's a great contribution to the field, in my opinion.

=====

Title: HSC70 coordinates COP9 signalosome and SCF ubiquitin ligase activity to enable a prompt stress response

The SCF (SKP1/CUL1/F-box protein) ubiquitin ligase complex plays a protective role against external stress, such as ultraviolet irradiation. The emergence of substrates activates SCF through neddylation, the covalent attachment of ubiquitin-like protein NEDD8 to CUL1. After substrate degradation, SCF is inactivated through deneddylation by COP9-signalosome (CSN), a solo enzyme that can deneddylate SCF. How the activity of CSN and SCF is coordinated within the cell is not fully understood. Here, we find that heat-shock cognate 70 (HSC70) chaperone coordinates SCF and CSN activation dependent on the neddylation status and substrate availability. Under basal conditions and low substrate availability HSC70 directly enhances CSN deneddylation activity, thereby reducing SCF activity. Under SCF-activated conditions HSC70 interacts with neddylated SCF and enhances its ubiquitination activity. The alternative interaction between HSC70 and CSN or neddylated SCF is regulated by

the presence or absence of SCF substrates. The knockdown of HSC70 decreases SCF-mediated substrate ubiquitination, resulting in vulnerability against ultraviolet irradiation. Our work demonstrates the pivotal role of HSC70 in the alternative activation of CSN deneddylation and SCF substrate ubiquitination, which enables a prompt stress response.-

Responses to the referees' comment

We thank the reviewers for the positive and constructive comments about our manuscript, "HSC70 alternatively regulates COP9 signalosome and SCF ubiquitin ligase for a prompt stress response (Manuscript ID; EMBOR-2024-59730V3)", by Nishimura et al. We performed additional experiments and revised the manuscript in response to the comments. We hope that we have responded adequately and that the revised manuscript is now suitable for publication.

*Please note that, in accordance with the editorial suggestions, we have modified several parts of the manuscript, including the title and abstract.

Response to referee 1

[Comment]

The revised manuscript by Nishimura et al. has been significantly improved with the addition of several new data including the use of HSC70 mutants that discriminate the role of HSC70 in CSN and SCF-Nedd8-substrate function.

[Response]

We appreciate your overall favorable evaluation, and the constructive comments provided. Your encouraging words motivate us to continue our work.

[Comment]

The *in vitro* data are now very convincing. However, in the absence of data regarding potential co-chaperones (the authors provide their reasoning for this) the presented data in cells are still based on looking the effect of HSC70 inhibition on protein stability, which as previously discussed are not very informative. If possible, this part will be strengthened potentially with the expression of HSC70 mutants such as the ATPase-defective mutant (HSC70 E175S) in HSC70 knockdown cells.

[Response]

Thank you for highlighting this important issue. Following your suggestion, we investigated whether ATPase-defective point mutant (HSC70 E175S) affects the half-life of MCL-1 in a *HSC70* knockdown cell. The expression of HSC70 E175S did not influence the protein half-life of MCL-1. This result aligns with *in vitro* results (Fig. 4D, E) showing that HSC70 E175S cannot activate the ubiquitination activity of SCF. Your suggestion has allowed us to further strengthen our *in vivo* findings. These results are included in the text and Fig. EV3E in the revised manuscript.

New Fig. EV3G

CHX chase assay of endogenous MCL-1 with or without transient FLAG-HSC70 E175S mutant expression. The graphs show the mean \pm s.e.m. (n=3, respectively; Welch's t-test). No significant difference was observed at each time point.

[Comment]

Also, the effect of HSC70 knockdown on substrate stability has to be presented in context of the effect of CSN3 inhibitors. The reviewer acknowledges that CSN inhibition could block SCF activity through sustained cullin hyper-neddylation but this has to be experimentally addressed and discussed.

[Response]

Thank you for your insightful comments. We analyzed the protein half-life of MCL-1 in both control and HSC70 knockdown cells under CSN5i-3 treatment. The CSN5i-3 treatment extended the protein half-life of MCL-1 to comparable levels in both conditions. Therefore, the observed difference in MCL-1 protein half-life between the control and HSC70 knockdown cells under untreated conditions (Fig. 6B) is likely attributable to the differential activity of SCF in these two cell types. These results are included and discussed in the text and Fig. EV3G in the revised manuscript.

New Fig. EV3E

CHX chase assay of endogenous MCL1 in HSC70 knockdown cells with pharmacological inhibition with CSN5i-3. The graphs show the mean \pm s.e.m. (n=3, respectively; Welch's t-test). No significant difference was observed at each time point.

Dr. Hidetaka Kioka
Osaka University
Department of Cardiovascular Medicine
Suita, Osaka 565-0871
Japan

Dear Dr. Kioka,

Thank you for the submission of your revised manuscript to EMBO Reports. I have editorially reviewed your additional experiments and implemented changes and am now very pleased to accept your manuscript for publication in the next available issue of EMBO reports. Thank you for your contribution to our journal.

Yours sincerely,
